# The circadian clock controls temporal and spatial patterns of floral development in sunflower

**Carine M Marshall[1], Veronica L Thompson[1], Nicky M Creux[1,2], Stacey L Harmer[1]***

[1]Department of Plant Biology, University of California, Davis, Davis, United States; [2]Department of Plant and Soil Sciences, FABI, Innovation Africa, University of Pretoria, Pretoria, South Africa

**Abstract** Biological rhythms are ubiquitous. They can be generated by circadian oscillators, which produce daily rhythms in physiology and behavior, as well as by developmental oscillators such as the segmentation clock, which periodically produces modular developmental units. Here, we show that the circadian clock controls the timing of late-stage floret development, or anthesis, in domesticated sunflowers. In these plants, up to thousands of individual florets are tightly packed onto a capitulum disk. While early floret development occurs continuously across capitula to generate iconic spiral phyllotaxy, during anthesis floret development occurs in discrete ring-like pseudowhorls with up to hundreds of florets undergoing simultaneous maturation. We demonstrate circadian regulation of floral organ growth and show that the effects of light on this process are time-of-day dependent. Delays in the phase of floral anthesis delay morning visits by pollinators, while disruption of circadian rhythms in floral organ development causes loss of pseudowhorl formation and large reductions in pollinator visits. We therefore show that the sunflower circadian clock acts in concert with environmental response pathways to tightly synchronize the anthesis of hundreds of florets each day, generating spatial patterns on the developing capitulum disk. This coordinated mass release of floral rewards at predictable times of day likely promotes pollinator visits and plant reproductive success.

*For correspondence: slharmer@ucdavis.edu

**Competing interest:** The authors declare that no competing interests exist.

## Editor's evaluation

This study successfully illustrates how the circadian clock spatiotemporally regulates unique developmental patterns of sunflower anthesis. It is generally true that the circadian clock underlies nearly all developmental processes in plants, but it could be surprising that the exquisite changes in flower shapes and growth in a disc structure are also shaped by the circadian clock. They also nicely put the emphasis on the relevance of the anthesis patterns in the ecological context in that the spatiotemporal coordination of flowering is essential for pollination and reproductive fitness.

## Introduction

From the earliest stages of cellular differentiation to reproductive maturity, spatio-temporal regulation of developmental transitions plays an important role in biological success. In some cases, internal temporal rhythms are used to spatially organize a body plan. For example, in vertebrate embryos a molecular 'segmentation clock' acts at regular intervals to divide the developing body axis into discrete blocks of cells that later differentiate into individual somites (*Richmond and Oates, 2012*). In plants, the regular spacing of lateral roots along the length of the primary root axis is controlled by a 'root clock' that generates a temporally oscillating developmental signal (*Xuan et al., 2020*). Thus

**eLife digest** Most organisms, from plants to insects and humans, anticipate the rise and set of the sun through an internal biological timekeeper, called the circadian clock. Plants like the common sunflower use this clock to open their flowers at dawn in time for the arrival of pollinating insects.

Sunflowers are composed of many individual flowers or florets, which are arranged in spirals around a centre following an age gradient: the oldest flowers are on the outside and youngest flowers on the inside. Each day, a ring of florets of different developmental ages coordinates their opening in a specific pattern over the day. For example, petals open at dawn, pollen is presented in the morning, and stigmas, the female organs that receive pollen, unfold in the afternoon. This pattern of flowering, or floret maturation, is repeated every day for five to ten days, creating daily rhythms of flowering across the sunflower head. Previously, it was unclear how florets within each flowering ring synchronize their flowering patterns to precise times during the day.

To find out more, Marshall et al. analysed time-lapse videos of sunflowers that were exposed to different day length and temperature conditions. Sunflowers opened a new floret ring every 24 hours, regardless of the length of the day. In all three day-length scenarios (short, middle, long), the development of the florets remained highly coordinated. Even flowers kept in the dark for up to four days were able to maintain the same daily growth rhythms. This persistence of daily rhythms in the absence of environmental cues suggests that the circadian clock regulates the genetic pathways that cause sunflowers to flower. However, when sunflowers whose circadian rhythms were delayed relative to the sun were placed out in a field, the sunflowers flowered later and thus attracted less pollinators.

Marshall et al. show that the circadian clock is important for regulating flowering patterns in sunflowers to ensure their successful pollination. A better understanding of the interplay between pollinators, flowering plants and their environment will provide more insight into how climate change may affect pollination efficiency. By identifying the genes and pathways underlying flowering patterns, it may be possible to develop breeds that flower at the optimal times of day to promote pollination. This could help mitigate the effects of climate change and declining populations of pollinators.

in both plants and animals, internal body clocks can generate serially repeated organs. These types of clocks are distinct from the circadian oscillator, the biological clock found in most eukaryotes and some prokaryotes (*Mosier and Hurley, 2021*). Circadian clocks coordinate development on daily and seasonal time scales. In both plants and animals, the circadian clock can integrate environmental cues such as day length and temperature to time the transition to reproductive stages to the appropriate season (*Ikegami and Yoshimura, 2012*; *Inoue et al., 2018*). On a diel scale, the circadian clock can coordinate developmental processes with environmental transitions associated with the earth's rotation, such as leaf movement in plants, eclosion in *Drosophila*, and conidiation in *Neurospora* (*Brady et al., 2021*; *Larrondo and Canessa, 2018*; *McClung, 2019*). However, a role for the circadian clock in the spatial patterning of development has not previously been reported.

Circadian clocks in eukaryotes are cell-autonomous systems made up of transcriptional-translational feedback loops that create rhythms of gene expression to regulate biological timing. Circadian rhythms, biological processes regulated by the circadian clock, meet several fundamental criteria (*Rensing and Ruoff, 2002*). First, they are stably entrained to environmental cues. Second, they persist in free-running environmental conditions. Third, circadian rhythms are temperature compensated such that they maintain a uniform period over a range of ambient temperatures, in contrast to the general rule that the rate of enzymatic processes changes with temperature. In addition, the circadian clock gates certain biological processes so that responsiveness to environmental cues is time-of-day dependent. For example, studies in humans show the timing of drug treatment can be critical to creating an effective therapeutic response (*Ruan et al., 2021*). A functioning circadian clock whose biological rhythms match that of the environment provides a fitness advantage for many organisms, including plants (*Dodd et al., 2005*), animals (*Miller et al., 2004*; *Emerson et al., 2008*; *Spoelstra et al., 2016*), bacteria (*Ouyang et al., 1998*), and fungi (*Koritala et al., 2020*).

One important way the circadian clock promotes fitness is by controlling timing of reproductive development, for example, by allowing organisms to distinguish between the long and short days of different seasons. In plants, the circadian system mediates the effects of day length during the

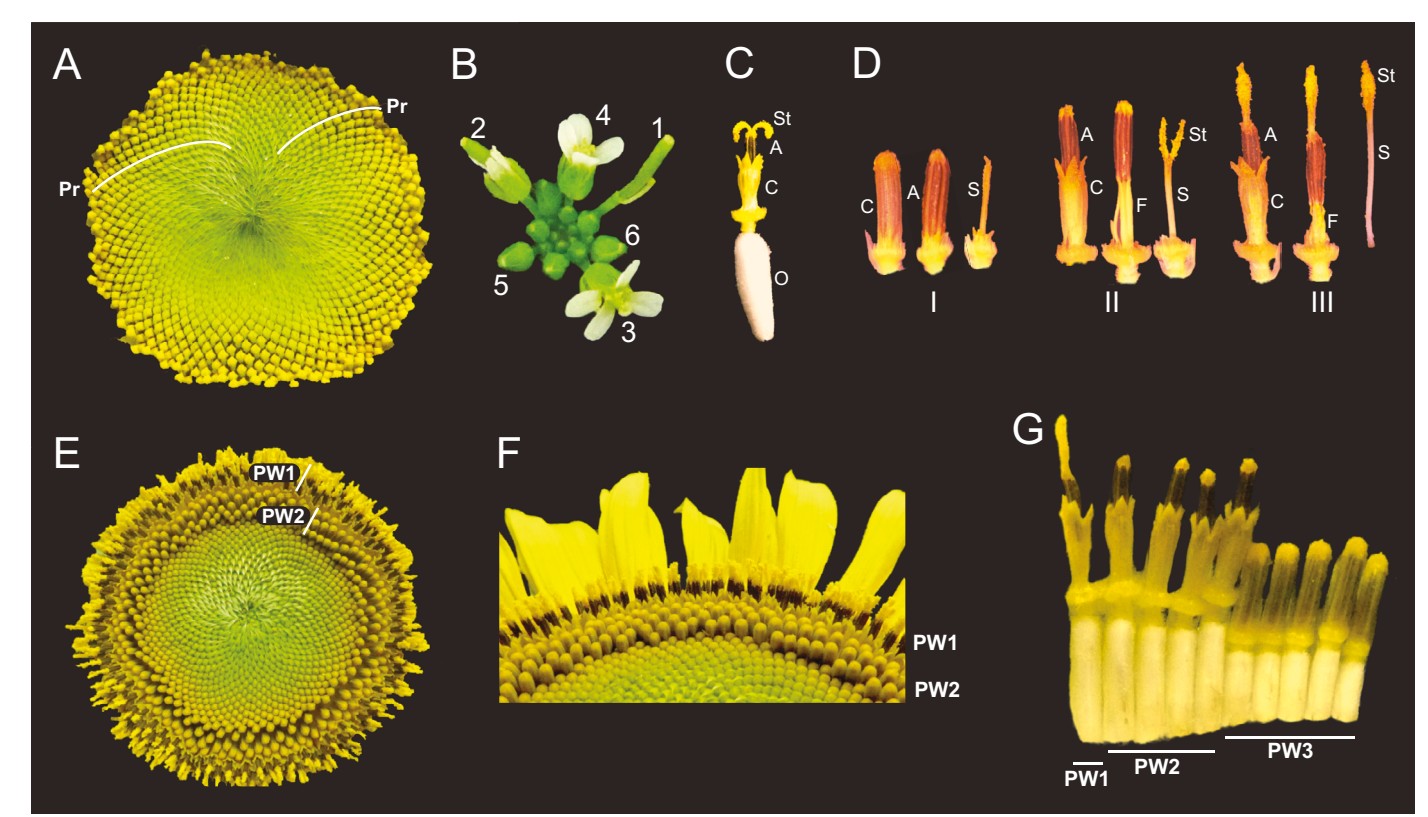

**Figure 1.** Architecture of the sunflower capitulum. (**A**) A sunflower capitulum before onset of anthesis, with obvious clockwise and counter-clockwise parastichies (Pr) of individual florets. (**B**) An *Arabidopsis thaliana* inflorescence also displays spiral phyllotaxy. Numbers indicate age of flowers. (**C**) One sunflower floret post-anthesis. Organs are abbreviated as ovary (O), corolla tube (C), anther tube (A), filaments (F), style (S), and stigma (St). Anther tubes and filaments collectively make the stamens. (**D**) The primary stages of floret development: (I) pre-anthesis immature, (II) early anthesis staminate, and (III) late anthesis pistillate. For each stage, the left-most floret is undissected, the corolla tube was removed from the middle floret, and the corolla and stamen were removed from the right-most floret. (**E**) A sunflower capitulum on day 2 of anthesis in which coordinated pseudowhorls (PW) of developing florets are apparent. (**F**) An angled view of (**E**). (**G**) A cross-section of (**E**); florets and ovaries along a single parastichy that belong to three different pseudowhorls. Images **C–G** were acquired at ZT 2. All plants grown in LD|25°C (16 hr:8 hr).

transition from vegetative to reproductive growth, usually referred to as flowering time (*Imaizumi and Kay, 2006*). The circadian clock also regulates daily aspects of floral development, for example, coordinating daily rhythms in scent emission and floral opening and closing with pollinator activity (*Fenske et al., 2015*; *Inoue et al., 2018*; *Muroya et al., 2021*; *Yon et al., 2017*). In addition, the circadian clock is implicated in the uniform eastward orientation observed in mature sunflower plants; this eastward orientation was shown to enhance both male and female reproductive fitness (*Atamian et al., 2016*; *Creux et al., 2021*).

Domesticated sunflower is an excellent model system to study the temporal regulation of floral development. Sunflowers are members of the Asteraceae, a highly successful family comprised of about 30,000 distinct species that are widely dispersed around the world (*Funk et al., 2009*). Asteraceae are characterized by the arrangement of individual florets together in a complex, compressed inflorescence called the capitulum. Individual florets are arranged on the capitulum disk in spiral patterns, a classic type of phyllotaxis observed in many plants (*Figure 1A and B*). These phyllotactic spirals, called parastichies, are generated by the precise specification of floral primordia, first at the outer rim and then moving into the center of the capitulum disk (*Zhang et al., 2021*). In domesticated sunflower, specification of the hundreds or even thousands of floret primordia found on a single capitulum takes approximately 10–14 days (*Marc and Palmer, 1981*; *Palmer and Steer, 1985*), followed approximately 20 days later by the onset of anthesis, or floral opening, of the first florets (*Lindström and Hernández, 2015*). Anthesis of all florets on a capitulum head typically occurs over 5–10 days (*Seiler, 2015*).

Intriguingly, the timing of anthesis does not follow the gradual and continuous pattern of development seen during specification of floret primordia on the developing capitulum (*Marc and Palmer, 1981*; *Zhang et al., 2021*). Instead, discrete developmental boundaries are observed with obvious rings of florets at distinct stages of anthesis (*Figure 1E and F*). Each ring, or pseudowhorl, undergoes anthesis on successive days. While previous studies examined the effects of hormones and environmental factors on the timing of development of florets within a pseudowhorl (*Baroncelli et al., 1990*; *Lobello et al., 2000*), the mechanisms governing the conversion of the continuous spiral pattern of development seen early in floral development (*Figure 1A*) to the ring-like patterns seen during anthesis (*Figure 1E*) have not been explored.

Here, we investigate the timing of floral organ development in various environmental conditions. We find that temperature-compensated daily rhythms in development of both male and female floral organs, and the formation of pseudowhorls, persist when flowers are maintained in constant environmental conditions in the absence of light. However, both daily rhythms in development and the formation of pseudowhorls are lost in constant light: late-stage floret maturation occurs continuously along parastichies, with negative impacts on pollinator visits. Importantly, the effects of light on floret development are time-of-day-dependent. Together, these data indicate that late-stage floret development is regulated by the circadian clock. Field experiments reveal that the timing of anthesis affects visits by pollinating insects, thus linking circadian control of floral maturation with plant reproductive success. We propose that pseudowhorls are produced in response to an external coincidence mechanism, in which developmentally competent florets undergo reproductive transitions only when they receive environmental cues at the appropriate time of day. Our data suggest a role for the circadian clock in the spatial as well as temporal control of sunflower development.

## Results

### During anthesis the pattern of development changes from continuous to discrete

The phyllotactic spirals of florets, or parastichies, on sunflower capitula represent continuous age-gradients with the first-specified florets on the outside edge of the capitulum (*Figure 1A*; *Marc and Palmer, 1981*). Each disc floret is an individual flower with an epigynous ovary that holds a corolla tube of fused petals (*Figure 1C*). Within the corolla tube is the fused anther tube, subtended by five filaments, collectively forming the stamen (*Figure 1C*). Finally, at the center of the floret is the style and stigma (*Figure 1C*), which we will together call the style. Although early floret development occurs continuously across the capitulum, anthesis does not. Local synchronization of floret development at this developmental stage converts the obvious spiral arrangement of florets on immature capitula into a ring-like pattern of pseudowhorls (*Figure 1A* vs. *Figure 1E*), marking a transition from a continuous to discrete pattern of development. This can be seen most clearly in the abrupt transition in ovary lengths between one pseudowhorl and the next (*Figure 1G*), with the longer ovaries of florets undergoing anthesis causing the protrusion of their corollas above those of immature florets (*Figure 1F*).

Previous studies described daily patterns in sunflower floret anthesis, with a single floret releasing pollen the day before the stigma of this floret is receptive to pollen (*Putt, 1940*; *Neff and Simpson, 1990*). Before anthesis all reproductive organs are enclosed within the corolla tube (*Figure 1D* – I). As anthesis begins, the corolla tubes of multiple florets within a pseudowhorl coordinately rise above the capitulum surface (*Figure 1E and F*) driven by ovary growth rather than expansion of the corolla (*Figures 1G and 2A*). Next, rapid elongation of anther filaments causes anther tube protrusion above the corolla (*Figure 1D* – II) (*Baroncelli et al., 1990*; *Lobello et al., 2000*). In this staminate stage of anthesis, pollen is released within the anther tube and is then pushed up and out by the rapid growth of the style through the anther tube soon after dawn (*Figure 1D* – III). Continued style elongation pushes the stigma beyond the anther tube to initiate the pistillate stage of anthesis (*Figure 1D* – III). This plunger-style mechanism of pollen presentation is an ancestral feature of the Asteraceae family (*Jeffrey, 2009*). The stigmas then unfold and become receptive to pollen the day after pollen release (*Figure 1C*). The temporal and spatial separation between pollen release and stigma presentation in sunflowers promotes cross-fertilization across the head (*Patterson, 2009*).

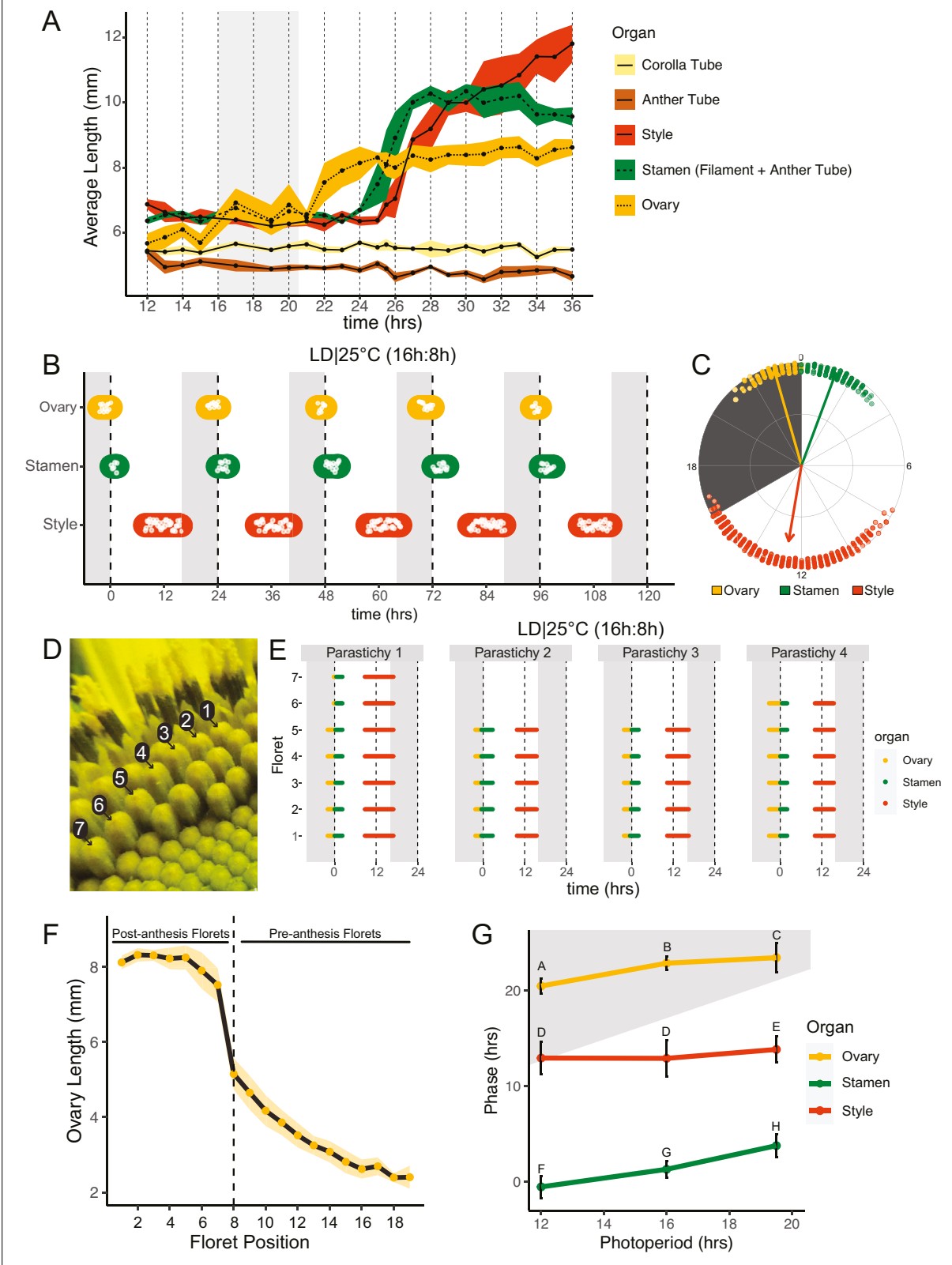

**Figure 2.** Pseudowhorls are due to coordinated daily rhythms in floret anthesis. (**A**) Growth kinetics of floral organs in a single pseudowhorl for plants grown in LD|25°C with dark from ZT 16–20.5. Corolla tube, anther tube, style+stigma, stamen (filament+anther tube), and ovary were measured. Black dots represent average lengths and ribbons represent standard deviation (n=7–15 florets per time point; n=1 biological replicate). (**B**) Timing of ovary, stamen, and style growth for florets on a capitulum in LD|25°C (16 hr:8 hr). Each white point represents time when organs of >5% of florets per

*Figure 2 continued on next page*

*Figure 2 continued*

pseudowhorl elongated (n=3 capitula); colored ovals group growth of organs within a pseudowhorl and are for ease of viewing rather than to convey statistical information. (**C**) The timing of active organ growth for pseudowhorls as seen in (**B**), superimposed on a 24 hr circular clock (n=3 capitula). Dots represent individual observations, arrow directions represent average phases, and arrow lengths represent precision of timing. (**D**) Numbered florets within a single parastichy. (**E**) Timing of active growth for all florets along a parastichy within pseudowhorl 2; plants grown in LD|25°C (16 hr:8 hr) (n=4 parastichies). (**F**) Lengths of ovaries along a parastichy at ZT 2; plants grown in LD|25°C (16 hr:8 hr). Position 8 represents the first floret in the next pseudowhorl (n=4 parastichies). Yellow dots indicate means and the ribbon represents standard deviation. (**G**) The average phases of active growth for ovaries, stamens, and styles in LD|25°C with dark from either ZT 12-24 (left), ZT 16-24 (middle), or ZT 16-20.5 (right); (n = 3-4 capitula). Error bars are standard deviation. Different letters indicate statistically significant differences in mean phases by one-way ANOVA with post-hoc Tukey HSD test (*P* < 0.001). For all graphs the period of dark exposure is shown with grey background.

The online version of this article includes the following figure supplement(s) for figure 2:

**Figure supplement 1.** Sunflower capitulum development in various environmental conditions.

**Figure supplement 2.** Sunflower anthesis is rhythmic in different photoperiodic conditions.

## Pseudowhorls are generated by synchronized, daily rhythms in floret development

To better understand the synchronization of anthesis within each pseudowhorl, we characterized the detailed temporal dynamics of late-stage floret development. Plants were maintained in light-dark cycles and constant temperature. Florets within a single pseudowhorl were dissected every 15–30 min over 24 hr and individual organ lengths measured. Ovary elongation was observed first, starting near the time of lights off and continuing for approximately 10 hr (*Figure 2A*). As previously described (*Baroncelli et al., 1990*; *Lobello et al., 2000*), we found that stamen elongation commenced approximately 8 hr after lights off, driven by an approximately 2.5× increase in anther filament length over 5 hr (*Figure 2A*). Anther tubes, like corolla tubes, did not elongate during this time frame. Stamens reached maximal length at time 28, due to filament elongation, and later shrank due to filament collapse (*Figure 2A*; *Figure 1D* – III). Styles initiated rapid elongation 1 hr after anther filaments, with growth continuing over 11 hr until they reached ~2× their original length and grew past the stamens. Thus, the obvious discrete phases of anthesis visible on whole capitula (*Figure 1E-G*) are driven primarily by the consecutive and highly coordinated elongation of ovaries, anther filaments, and styles within each pseudowhorl.

We next investigated the effects of different environmental conditions on the dynamics of floret development. We generated time-lapse movies of sunflowers undergoing anthesis taken at 15 min intervals and scored observable patterns of growth as proxies for the detailed organ measurements described above. We first examined growth kinetics in capitula undergoing anthesis in long-day conditions reminiscent of a summer day (16 hr light, 8 hr dark, and constant 25°C (LD|25°C [16 hr:8 hr])). A new pseudowhorl of florets underwent anthesis every 24 hr, with ovary swelling initiating in the late night, stamen elongation visible from late night to soon after dawn, and conspicuous style elongation occurring in the latter part of the day and early night (*Figure 2B*; *Figure 2—figure supplement 1A*; *Table 1*; *Video 1*). Times of ovary, stamen, and style growth were highly coordinated across pseudowhorls and flowers (*Figure 2C*; *Video 1*). The mean midpoint of visible ovary growth occurred before dawn at zeitgeber time (ZT) 22.9 (with the time of lights on defined as ZT 0), followed by the mean midpoint of stamen growth at ZT 1.3, and the mean midpoint of style growth before dusk at ZT 12.6 (*Figure 2B and C*; *Table 1*). Similar experiments were carried out for plants exposed to shorter (LD|25°C [12 hr:12 hr]) or longer (LD|25°C [19.5 hr:4.5 hr]) photoperiods. Similar daily rhythms of anthesis were observed in all three conditions, with the phases of organ growth relative to dawn changing only slightly with different night lengths (*Figure 2G*; *Figure 2—figure supplement 2*). In all three photoperiod conditions, the timing of floret development within pseudowhorls was highly coordinated (*Figure 2B*; *Figure 2—figure supplement 2*) with a stable phase relationship between all organs. Such phase stability across different photoperiods is often seen for circadian-regulated processes (*Wang et al., 2022*).

This high degree of developmental coordination within each pseudowhorl is quite remarkable given that the initiation of floret primordia along each phyllotactic spiral of a parastichy occurs continuously over many days (*Zhang et al., 2021*). We therefore examined the timing of ovary, stamen, and style growth within a pseudowhorl along separate parastichies (*Figure 2D and E*). While the number of florets per parastichy within a pseudowhorl varied, growth of the three different organs occurred at

**Table 1.** Circadian parameters for anthesis in different environmental conditions.

| Condition | Organ | n | Mean period (hr) | St. dev. | ZT window for period analysis (hr) | Mean phase (hr) | St. dev. | Phase analysis window (pseudowhorls) |
|---|---|---|---|---|---|---|---|---|
| LD\|25°C (16 hr:8 hr) | Ovary | 3 | 24.2 | 0.4 | | 22.9 | 0.7 | |
| | Stamen | 3 | 24.4 | 0.3 | | 1.3 | 0.8 | |
| | Style | 3 | 24.4 | 0.1 | 21–137 | 12.6 | 2.1 | 1–5 |
| LD\|25°C (12 hr:12 hr) | Ovary | 4 | 25.2 | 0.9 | | 20.5 | 0.8 | |
| | Stamen | 4 | 25.0 | 0.7 | | 23.5 | 1.5 | |
| | Style | 4 | 23.7 | 1.5 | 17–136 | 13.1 | 1.7 | 3–6 |
| LD\|25°C (19.5 hr:4.5 hr) | Ovary | 3 | 23.5 | 0.6 | | 19.9 | 1.5 | |
| | Stamen | 3 | 23.2 | 0.1 | | 0.3 | 1.2 | |
| | Style | 3 | 23.8 | 1.5 | 18–110 | 10.3 | 1.4 | 3–6 |
| DD\|18°C | Ovary | 3 | 22.5 | 0.2 | | 22.4 | 2.3 | |
| | Stamen | 3 | 22.3 | 0.4 | 17.5–81 | 5.1 | 2.6 | 2–4 |
| DD\|25°C | Ovary | 4 | 21.3 | 0.6 | | 18.4 | 3.5 | |
| | Stamen | 4 | 21.8 | 0.2 | 17–98 | 0.0 | 3.7 | 2–5 |
| DD\|30°C | Ovary | 3 | 21.9 | 1.1 | | 20.4 | 3.6 | |
| | Stamen | 3 | 22.9 | 0.5 | 18–80 | 1.2 | 2.1 | 2–4 |
| LL\|30–18°C (16 hr:8 hr) | Ovary | 3 | 22.8 | 0.2 | 15–97 | 20.2 | 3.2 | 2–5 |
| DD\|30–18°C (16 hr:8 hr) | Ovary | 3 | 23.4 | 0.4 | | 19.0 | 1.3 | |
| | Stamen | 3 | 23.5 | 0.3 | | 23.3 | 1.2 | |
| | Start Style | 3 | 21.7 | 2.2 | 17–98 | 2.4 | 0.6 | 3–5 |
| LDLD\|25°C (6 hr:6 hr:4 hr:8 hr) | Ovary | 3 | 24.2 | 0.5 | | 21.2 | 2.3 | |
| | Stamen | 3 | 24.0 | 0.1 | | 2.1 | 0.9 | |
| | Style | 3 | 23.9 | 0.1 | 40–136 | 13.5 | 1.8 | 3–6 |

almost the identical time for all florets (*Figure 2E*). Ovary lengths were also measured along individual parastichies at ZT 2, a time when ovary growth within a pseudowhorl is complete (*Figure 2A*). The lengths of ovaries within a pseudowhorl along a parastichy were similar to each other (*Figures 1G and 2F*), but there was a large step-down in length compared to ovaries of pre-anthesis florets in the same parastichy. Previous studies showed that the large changes in ovary size and filament length during anthesis are due to rapid cell expansion (*Lindström et al., 2007*; *Lindström and Hernández, 2015*; *Lobello et al., 2000*). The remarkable synchronization of cell elongation and organ growth during anthesis, despite the age difference of florets along each parastichy, suggests some master regulator coordinates anthesis within each pseudowhorl.

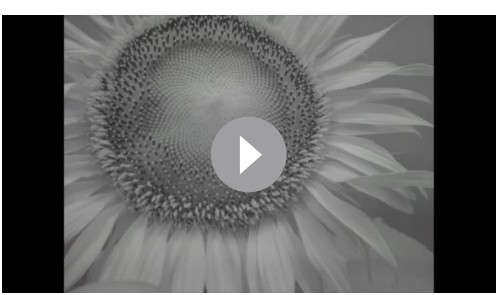

**Video 1.** Anthesis on a capitulum maintained in light/dark cycles (16 hr light, 8 hr dark) at a constant 25°C.
https://elifesciences.org/articles/80984/figures#video1

## Daily rhythms in floret development persist in constant dark conditions and are temperature-compensated

The precise daily timing of sunflower anthesis suggested that the circadian clock might be

involved in this process. To test this, we examined the development of sunflower capitula maintained in constant dark and temperature. Rhythms in ovary and stamen elongation persisted in these free-running conditions, with a new pseudowhorl of florets undergoing anthesis every ~24 hr with phases similar to those seen in light/dark cycles (*Figure 3A and B*; *Video 2*). Scoring of individual florets along a parastichy confirmed coordinated floret organ growth within a pseudowhorl (*Figure 3C*), even after 4 days in DD|25°C (*Figure 3—figure supplement 1*). We next examined ovary lengths among the florets along individual parastichies. As in photoperiodic conditions, ovary lengths between pseudow-horls were clearly distinct (*Figure 3E and F*). Thus, daily rhythms in anthesis and the formation of pseudowhorls persisted in the absence of environmental cues, suggesting regulation by the circadian clock.

As seen for ovaries and stamens, the start of visible style elongation within each pseudowhorl occurred at the expected times and was highly coordinated for the first 2 days in constant darkness (*Figure 3A and C*; *Figure 3—figure supplement 1*; *Video 2*). However, style elongation rates were very slow in DD|25°C (*Figure 3—figure supplement 2*) such that style growth in one pseudowhorl continued even after florets in the subsequent pseudowhorl began anthesis. This resulted in almost continuous style elongation across the capitulum. The immediate slowing of style but not ovary or stamen elongation upon transfer to constant darkness (*Figure 3—figure supplement 2*) suggests that different regulatory pathways control growth of these organs. However, the daily rhythms in initiation of growth of all three types of organs in constant darkness suggest a general role for the circadian clock in the control of anthesis.

Circadian-regulated processes are characterized by their relative insensitivity to temperature vari-ation across a physiologically relevant range (*Greenham and McClung, 2015*). We therefore deter-mined the free-running periods for ovary and stamen developmental rhythms in sunflowers maintained in constant darkness at 18°C, 25°C, or 30°C. Across the three different growth conditions, the esti-mated periods for cycles of ovary development were not statistically different while the estimated period for stamen development was slightly but significantly longer at 30°C, suggesting modest over-compensation (*Figure 3D*; *Table 1*). However, within each temperature, the estimated periods of ovary and stamen growth were not significantly different (*Figure 3D*). The rhythmic internal processes that regulate timing of floret anthesis are therefore temperature-compensated, further suggesting regulation by the circadian clock.

## The circadian clock gates floral responsiveness to dark cues

Environmental influences on clock-regulated processes are often 'gated' by the circadian clock, meaning that the same stimulus can evoke different responses when applied at different times of day (*Wenden et al., 2011*). We therefore examined whether the effects of light and darkness on sunflower anthesis are time-of-day dependent by applying a short 4.5 hr dark period at different times of the subjective day or night (*Figure 4A*) and assessing the effects on the spatial and temporal regulation of floret development. When the onset of darkness occurred during the subjective night, anthesis proceeded in the expected pattern with minor phase adjustments (*Figures 2A and 4C*; *Figure 4—figure supplement 1*). However, when the onset of darkness occurred during the subjective day, different effects were observed for ovaries compared to stamens and styles. In pseudowhorl 2, the first pseudowhorl undergoing development in the new light conditions, ovary elongation began at the expected time but with a slow growth rate that continued until the next period of darkness (*Figure 4B and C*; *Figure 4—figure supplement 1*). In contrast, development of stamens and styles did not initiate until ovary growth within pseudowhorl 2 was complete. As a result, dark treatments during the subjective day caused a considerable lag in anthesis (*Figure 4C*; *Figure 4—figure supplement 1*). After 2 days in each new photoperiodic condition, the phases of ovary, stamen, and style elonga-tion shifted such that maturation of all floret organs occurred with stable phase relationships to the new times of dusk and dawn (*Figure 4—figure supplement 1A–G, I*). This rapid shift in the phase of anthesis suggests that re-entrainment of the circadian clock may occur quickly in sunflower capitula.

We further characterized anthesis in the ZT 14–18.5 condition, in which darkness occurred 2 hr earlier than in the original entraining conditions. In this condition, there was a 42.9 hr difference between start of pseudowhorl 2 ovary growth and the start of pseudowhorl 3 ovary growth, compared to the 22.8 hr difference seen in the ZT 16–20.5 dark condition (*Figure 4B and C*; *Figure 4—figure supple-ment 1J*; *Table 2*). Surprisingly, this large delay in anthesis did not cause two pseudowhorls to develop

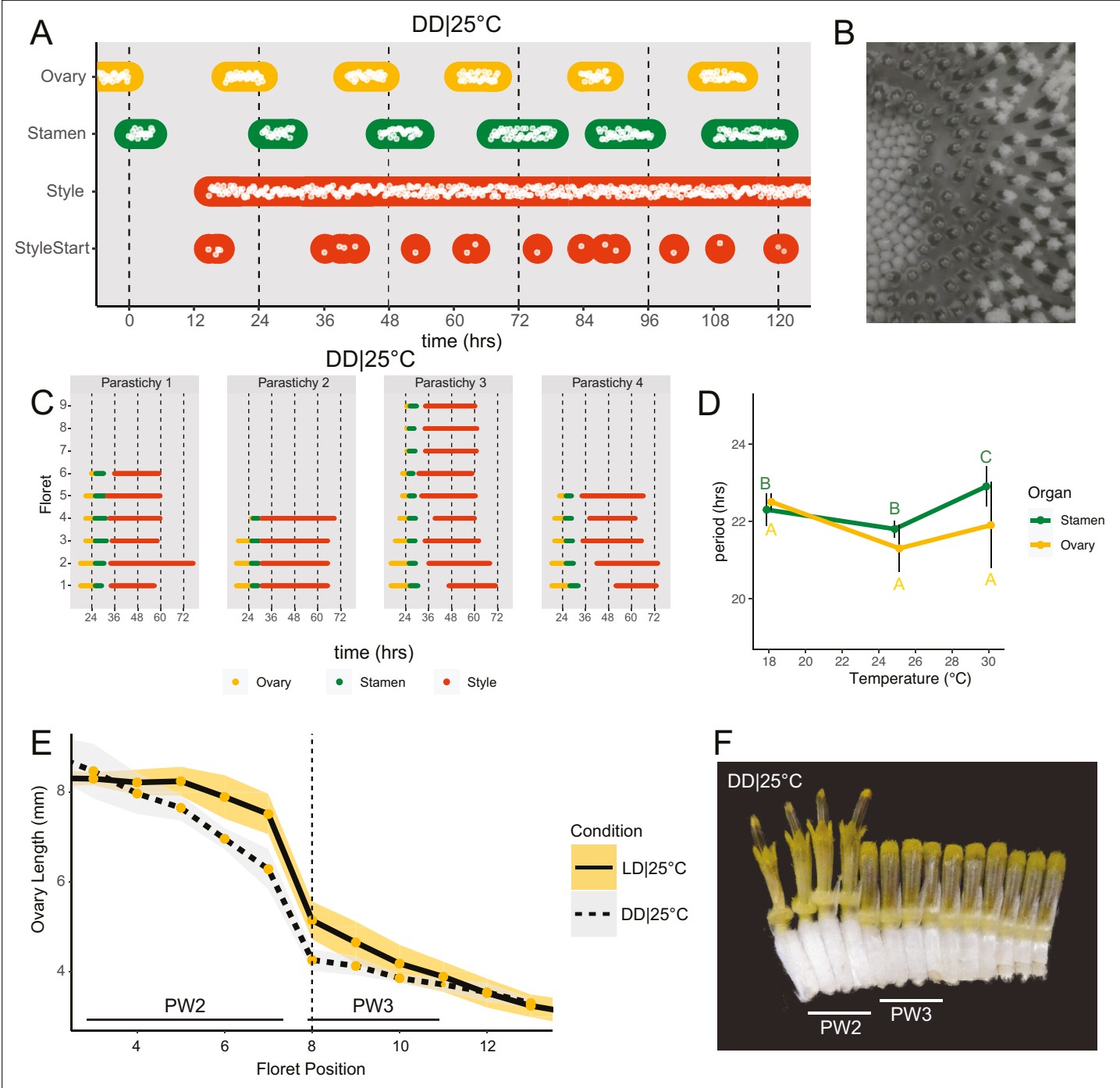

**Figure 3.** Daily rhythms in floret anthesis are regulated by the circadian clock. (**A**) Timing of ovary, stamen, style, and start-of-style growth for florets in DD|25°C (n=4 capitula). (**B**) Sunflower capitulum in DD|25°C at zeitgeber time (ZT) 50, showing coordinated floret anthesis. (**C**) Timing of active growth for all florets per parastichy within pseudowhorl 2; DD|25°C (n=4 parastichies). (**D**) Average periods (hr) of rhythmic ovary and stamen growth in DD|18°C (n=3), DD|25°C (n=4), or DD|30°C (n=3 capitula). Error bars represent standard deviation. Different letters indicate statistically significant differences by one-way ANOVA with post hoc Tukey HSD test (p<0.001), with yellow letters for differences within ovaries and green letters for differences within stamens. Periods of stamen and ovary growth are not statistically different within any temperature treatment. (**E**) Lengths of ovaries along a parastichy for sunflowers grown in either LD|25°C (16 hr:8 hr) (n=4 parastichies) or DD|25°C (n=8 parastichies) at ZT 26, with position 8 the first ovary in the next pseudowhorl. Yellow dots indicate means and the ribbons standard deviation. (**F**) A cross-section of a capitulum grown in DD|25°C; florets and ovaries along a single parastichy that belong to two different pseudowhorls. Photograph taken at ZT 26. Plots are as described for *Figure 2*.

The online version of this article includes the following figure supplement(s) for figure 3:

*Figure 3 continued on next page*

**Figure supplement 1.** Pseudowhorl coordination persists for multiple days in constant dark free-running conditions.

**Figure supplement 2.** Rates of floral organ maturation are dependent on environmental conditions.

simultaneously: the average number of new florets undergoing anthesis within the new pseudowhorl was the same across the different light treatments (*Figure 4—figure supplement 2*). Close inspection of the timing of development of individual florets along parastichies within pseudowhorl 2 revealed desynchronization of anthesis in the ZT 14–18.5 dark condition (*Figure 4E*). In contrast, dark treatment from ZT 16 to 20.5 promoted the highly coordinated pattern of anthesis seen in LD|25°C (16 hr:8 hr) conditions (*Figures 2E and 4D*). The large difference in maturation patterns depending upon whether lights were turned off at ZT 14 or at ZT 16 demonstrates a time-of-day dependent response to dark cues, a further indication of circadian clock regulation of this process.

Many circadian-regulated processes exhibit frequency demultiplication; that is, they maintain 24 hr rhythms even when subjected to light/dark cycles with a cycle length close to half that of the endogenous period (*Roenneberg et al., 2005*). We therefore examined whether multiple periods of darkness during a 24 hr period would affect the timing of floret anthesis within pseudowhorls. We first exposed sunflowers to four equal periods of darkness and light every 24 hr, LDLD|25°C (6 hr:6 hr:6 hr:6 hr). One pseudowhorl of florets underwent anthesis every 24 hr, but florets within each pseudowhorl were poorly coordinated (*Figure 5—figure supplement 1B*).

We next changed the proportion of darkness in a 24 hr period to a 6 hr period of darkness during the subjective day followed by an 8 hr period of darkness during the subjective night (LDLD|25°C [6 hr:6 hr:4 hr:8 hr]). This combination restored coordination of 24 hr rhythms of floret development within each pseudowhorl (*Figure 5A*). In addition, phases of organ development were very similar to those observed in LD|25°C (16 hr:8 hr) entrainment conditions (*Figure 5*; *Figure 5—figure supplement 1A*; *Figure 5—figure supplement 2B*; *Table 1*). Thus, the 6 hr dark period during the subjective day had no effect on the timing of anthesis, further suggesting circadian clock gating of responses to darkness.

## Pseudowhorl formation depends upon coordinated daily rhythms of anthesis

Previous studies show that sunflower anther filament elongation is completely inhibited and style elongation is reduced in constant light conditions (*Baroncelli et al., 1990*; *Lobello et al., 2000*). To investigate whether constant light also affected other aspects of flower development, we entrained sunflowers in LD|25°C (16 hr:8 hr) and then transitioned them to constant light and temperature (LL|25°C) on the first day of anthesis. As expected, no stamen growth was observed (*Figure 6A and F*; *Figure 6—figure supplement 1A*; *Video 3*). Ovaries and styles did elongate, but initiation occurred continuously across the capitulum. Instead of the coordinated floret anthesis seen along a parastichy of florets in LD|25°C (16 hr:8 hr) (*Figure 2E*; *Video 1*), in LL|25°C florets showed a continuous, age-dependent pattern of anthesis (*Figure 6A and B*; *Figure 6—figure supplement 1B*; *Video 3*). Consistent with continuous rather than discrete developmental transitions across the capitulum in constant light, ovary length along a parastichy showed a size gradient dependent on the age of the floret (*Figure 6E and F*). Thus, in constant light conditions the timing of anthesis mirrors the continuous pattern of floret specification seen very early in capitulum development. This loss of circadian coordination of anthesis causes a loss of coordination of florets into pseudowhorls (*Figure 6F*).

We next assessed the effects of temperature cycles on anthesis in plants maintained in constant light or dark. In constant darkness with temperature cycles of 16 hr at 30°C and 8 hr of 18°C (DD|30–18°C [16 hr:8 hr]), rhythms in the onset of anthesis were similar to those seen without temperature cycles albeit with enhanced

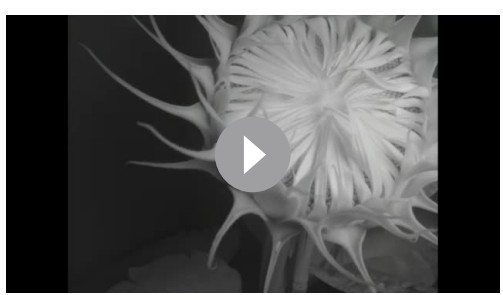

**Video 2.** Anthesis on a capitulum maintained in constant darkness and temperature (25°C).
https://elifesciences.org/articles/80984/figures#video2

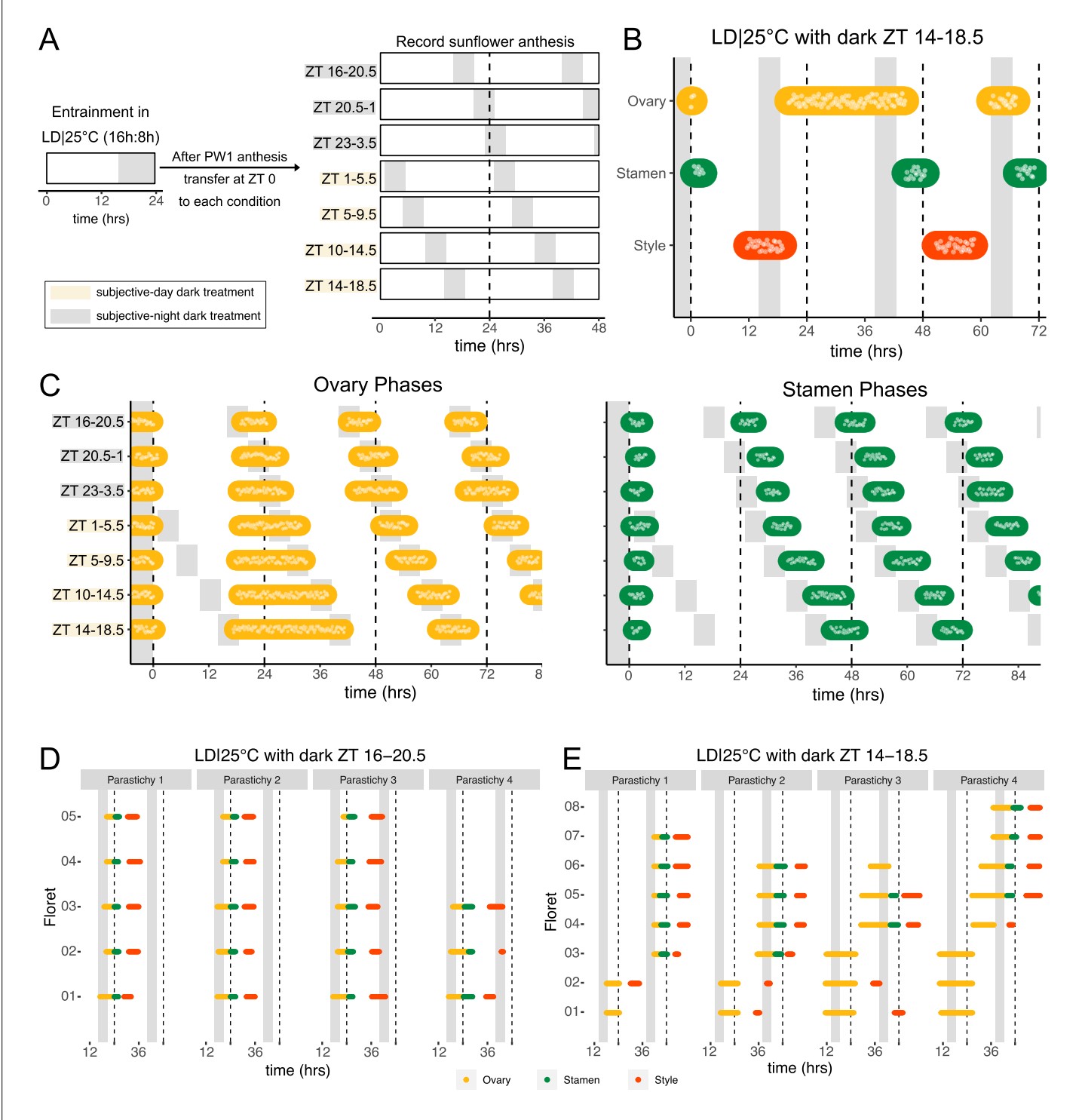

**Figure 4.** Normal rhythms of anthesis are maintained with darkness during the subjective night but not during the subjective day. (**A**) Schematic representation of experiment. All plants were maintained at 25°C with 4.5 hr of darkness provided at the times indicated with gray boxes. (**B**) Timing of ovary, stamen, and style growth with dark during zeitgeber time (ZT) 14–18.5 (n=4 capitula). (**C**) Timing of ovary (left) and stamen (right) growth with darkness provided during the indicated times (n=3–4 capitula per condition). (**D, E**) Timing of active growth for all florets along a parastichy within pseudowhorl 2 of sunflowers with dark during (**D**) ZT 16–20.5 or (**E**) ZT 14–18.5 (n=4 parastichies for each condition). Plots are as described for *Figure 2*.

The online version of this article includes the following figure supplement(s) for figure 4:

**Figure supplement 1.** Phase of organ growth resets rapidly to new entrainment conditions.

**Figure supplement 2.** Number of florets in a pseudowhorl not altered after delay in anthesis.

**Table 2.** Time between initiation of PW2 and PW3.

| Period of darkness (ZT) | Mean time between PW2 and PW3 (hr) | St. dev. |
| --- | --- | --- |
| 16–20.5 | 22.8 | 0.76 |
| 20.5–1 | 25.6 | 0.53 |
| 23–3.5 | 25.4 | 2.13 |
| 1–5.5 | 30.6 | 0.29 |
| 5–9.5 | 33.5 | 2.47 |
| 10–14.5 | 38.3 | 0.66 |
| 14–18.5 | 42.9 | 1.00 |

coordination of initiation of style elongation in each pseudowhorl (*Figure 2—figure supplement 1I*; *Figure 6—figure supplement 2A*; *Table 1*). We next investigated whether temperature cycles could rescue daily rhythms of floret anthesis in constant light (LL|30–18°C [16 hr:8 hr]). Although anther filament growth was not restored, the addition of thermocycles to constant light coordinated the onset of ovary and style growth so that pseudowhorls formed (*Figure 6C*; *Figure 2—figure supplement 1H*; *Figure 6—figure supplement 1C*; *Table 1*). Scoring of individual florets along a parastichy also showed that anthesis was coordinated into discrete pseudowhorls in constant light with thermocycles (*Figure 6D*; *Figure 6—figure supplement 2D*). As expected, this generated a discontinuous change in ovary length along parastichies from post-anthesis florets in a pseudowhorl to pre-anthesis florets (*Figure 6E*). Thus, temperature cycles restore some daily anthesis rhythms and thereby rescue pseudowhorl formation in constant light.

## Light-induced changes in anthesis patterns disrupt pollinator visits

We wished to investigate whether alterations in the patterns or timing of anthesis affect plant-pollinator interactions. To examine the ability of inflorescences undergoing uncoordinated anthesis (*Figure 6*) to attract pollinators, we transitioned plants 1 day before the start of anthesis to growth chambers with either constant light and temperature (LL|28°C) conditions or cycling temperature and light conditions similar to those found in the field. After 24 hr in these conditions, (*Figure 7A*), plants were transferred

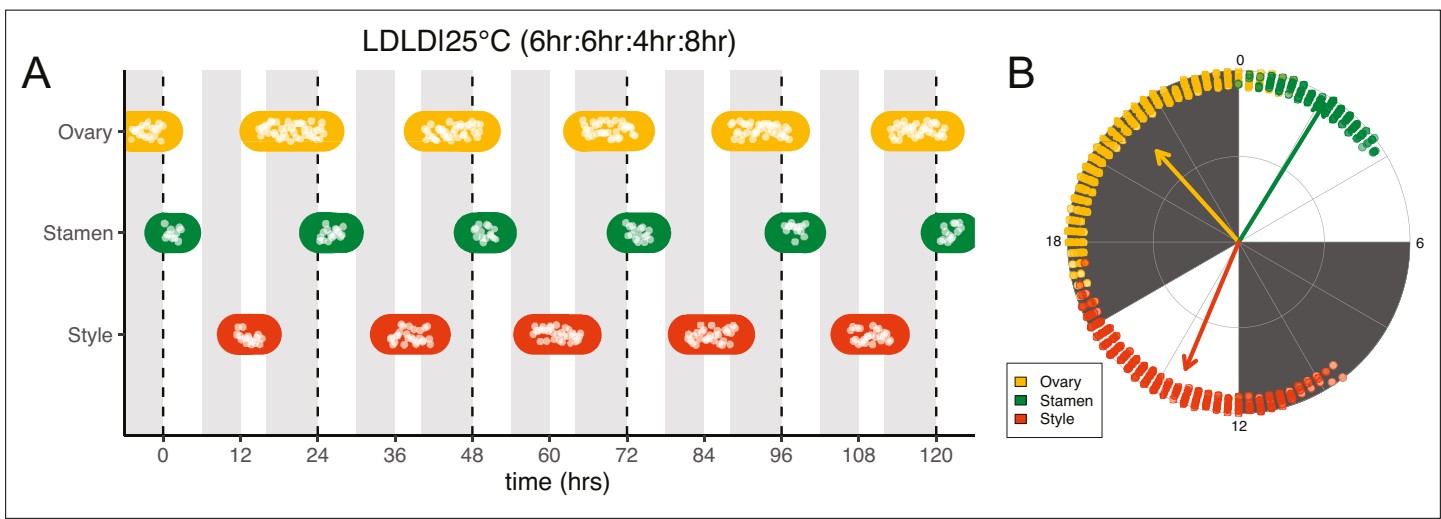

**Figure 5.** Frequency demultiplication of sunflower anthesis rhythms. (**A**) Timing of ovary, stamen, and style growth for florets on a capitulum with two cycles of light and dark each day, namely 6 hr L, 6 hr D, 4 hr L, 8 hr D (n=3 capitula). (**B**) The timing of active organ growth for pseudowhorls as seen in (**A**), superimposed on a 24 hr circular clock (n=3 capitula). Plots are as described for *Figure 2*.

The online version of this article includes the following figure supplement(s) for figure 5:

**Figure supplement 1.** Frequency demultiplication in anthesis requires a dominant night.

**Figure supplement 2.** Relative phases of floret development in different environmental conditions.

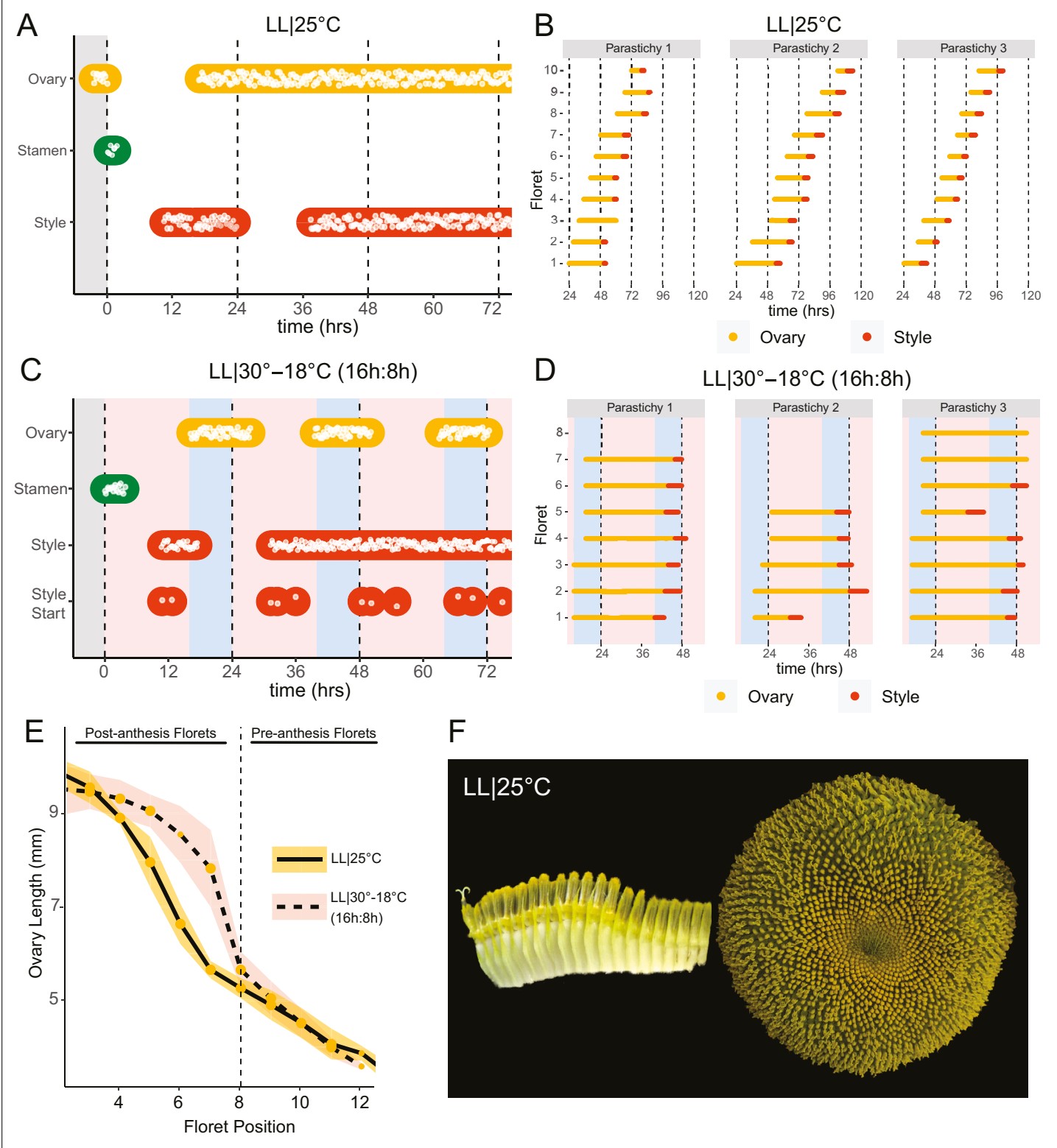

**Figure 6.** Constant light disrupts coordination of anthesis and pseudowhorl formation. (**A, C**) Timing of ovary, stamen, and style growth for florets on a capitulum in (**A**) LL|25°C (n=3 capitula), and (**C**) LL|30–18°C (16 hr:8 hr) (n=3 capitula). (**B, D**) Timing of active growth 24 hr after transfer to (**B**) LL|25°C, 10 consecutive florets measured along each parastichy (n=3 parastichies), and (**D**) LL|30–18°C (16 hr:8 hr) for all florets per parastichy in pseudowhorl 2 (n=3 parastichies). (**E**) Lengths of ovaries along a parastichy for sunflowers grown in either LL|25°C (n=4 parastichies) or LL|30–18°C (16 hr:8 hr) (n=5 parastichies) at zeitgeber time (ZT) 26, with position 8 the first ovary in the next pseudowhorl. (**F**) Florets along a parastichy at ZT 26 and the comparable

*Figure 6 continued on next page*

*Figure 6 continued*

sunflower capitulum at ZT 120; plants grown in LL|25°C. Periods of constant light at 30°C (pink background), 25°C (white background), and 18°C (blue background) are indicated; plots are otherwise as described for *Figure 2*.

The online version of this article includes the following figure supplement(s) for figure 6:

**Figure supplement 1.** Sunflower anthesis is impaired in constant light.

**Figure supplement 2.** Thermocycles in constant dark strengthen coordination of rhythms of floret development.

at dawn to a field site and time-lapse images were taken to monitor pollinator visits and floret development. As expected, plants transitioned from constant conditions did not undergo anther elongation or release pollen (*Figure 7B–D*). These plants attracted many fewer pollinators at all time points assessed (*Figure 7B*; *Table 3*) and significantly fewer pollinators overall (*Figure 7C*; *Table 3*). Thus light-mediated disruption of anthesis negatively affected floral interactions with pollinators.

We next examined whether changes in the timing of anthesis, rather than its disruption, affect plant-insect interactions. We therefore monitored the visits of pollinators to plants that were entrained in delayed light conditions in growth chambers and then transferred to the field. While control plants were entrained in phase with local environmental conditions, experimental plants were subjected to the same cycling conditions but with a 3 hr phase delay relative to the controls (*Figure 7E*). During anthesis of pseudowhorls 2–3, plants were transferred from growth chambers to the field before local dawn and monitored for 2 consecutive days. The jet lag plants exhibited a 1.3 hr delay in pollen release relative to the controls (*Figure 7F and H*; *Table 3*), a smaller phase delay than expected. Since warmer temperatures promote style elongation and pollen release (*Creux et al., 2021*), this phase advance relative to entraining conditions may be due to solar warming of the east-facing flowers. Pollinator visits closely tracked the timing of pollen release, with insects observed on jet lagged inflorescences approximately 1.5 hr later than on control inflorescences (*Figure 7F*; *Table 3*). The overall number of pollinator visits to the jet lag plants was significantly lower than those to controls (*Figure 7G*; *Table 3*). We continued to monitor these plants on their second day in the field. Consistent with the rapid re-entrainment of sunflowers in the growth chamber (*Figure 4*), pollen release of the jet lag and control plants was coordinated with local dawn and there was no significant difference in pollinator visits to the experimental and control plants (*Figure 7—figure supplement 1*). Since we previously showed that greater numbers of morning pollinator visits to sunflower inflorescences are correlated with greater siring success (*Creux et al., 2021*), these data implicate circadian control of anthesis in plant reproductive fitness.

## Discussion

Many aspects of floral physiology, including flower opening and scent production, vary across a day (*Bloch et al., 2017*). In a few cases, such as petal opening and closing in *Kalanchoë blössfedliana* and flower movements in *Nicotiana attenuata*, the circadian clock has been shown to regulate these diel rhythms (*Bünsow, 1953*; *Yon et al., 2016*). Here, we show that daily rhythms in the development of floral organs in sunflower persist in constant environmental conditions, are temperature-compensated, and exhibit a time-of-day specific response to environmental cues (*Figure 3A and D*; *Figure 4*; *Figure 5*). We therefore conclude that the circadian clock controls floral development in sunflower to ensure release of pollen soon after dawn and the emergence of styles and stigmas late in the day. Although style elongation is largely complete near the end of the first day of floret anthesis (*Baroncelli et al., 1990*; *Figure 2A*), stigmas do not become receptive to pollen until the following morning (*Putt, 1940*; *Neff and Simpson, 1990*). This temporal separation between release of pollen and stigma

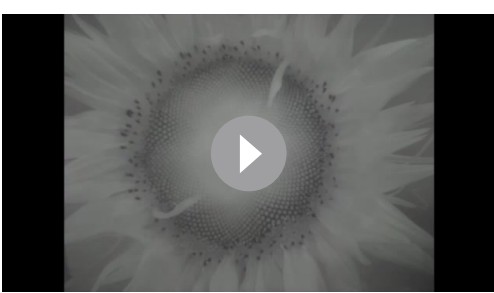

**Video 3.** Anthesis on a capitulum maintained in constant light and temperature (25°C).
https://elifesciences.org/articles/80984/figures#video3

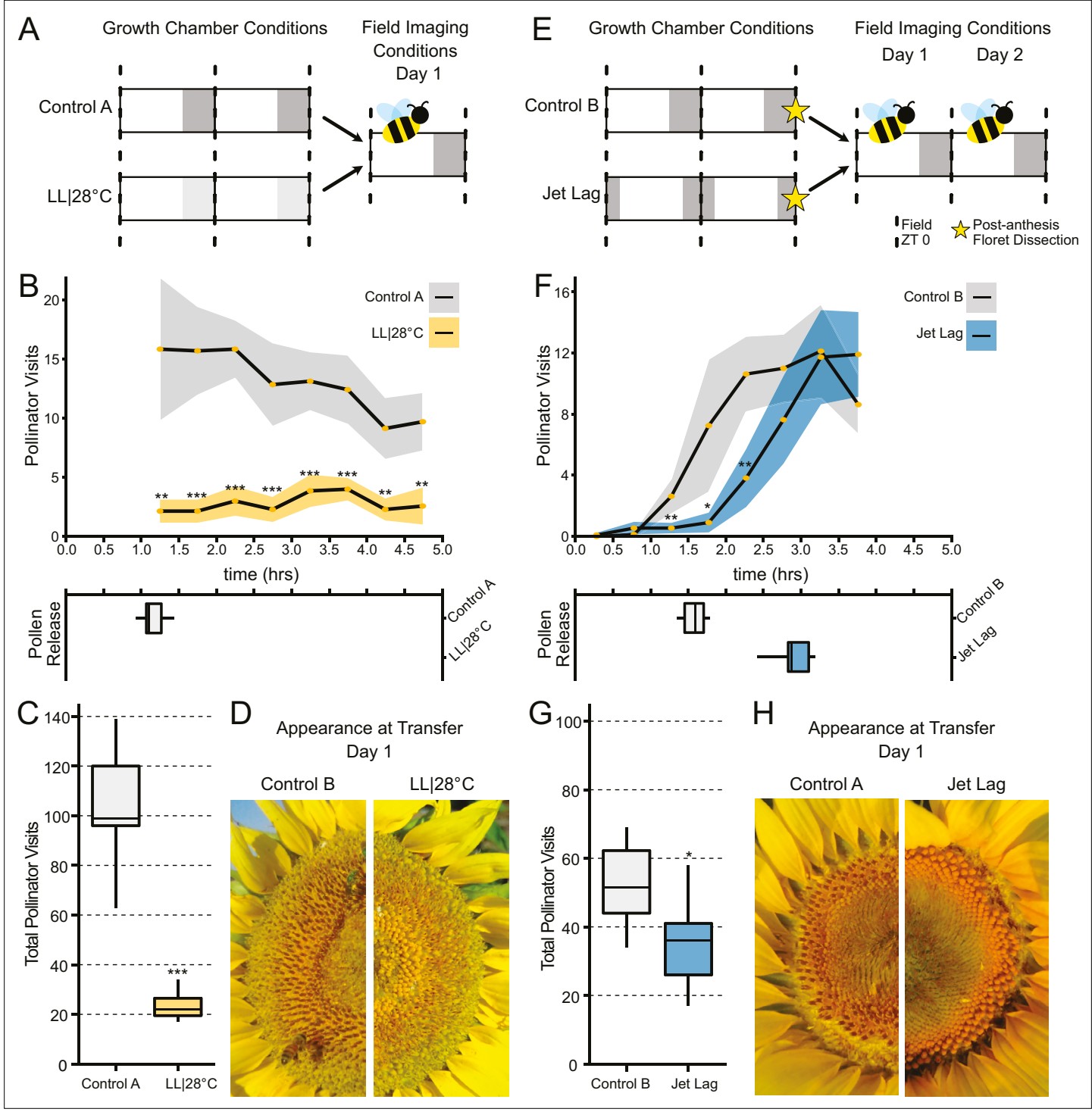

**Figure 7.** Effect of the circadian clock and discrete anthesis patterns on pollinator visits to sunflowers. (**A, E**) Schematic representation of experiment. Dark gray boxes indicate timing of darkness, light gray boxes indicate subjective night. Arrows show time when plants were transitioned to the field and stars indicate dissections of post-anthesis (open) florets. Control plants were maintained in conditions consistent with the local environment in Davis, CA. (**B, F**) Timing of pollinator visits and pollen presentation. Pollinator visits per 30 min bin were quantified starting immediately upon transfer to the field. (**B**) Control A (n=7 capitula), LL|28°C (n=7 capitula). (**F**) Control B (n=8 capitula), jet lag (n=11 capitula). Yellow dots represent means, ribbons represent 95% confidence intervals, stars indicate statistically significant differences by one-way ANOVA test (***p<0.001, **p<0.01, *p<0.05). (**C, G**) Total pollinator visits over the period quantified in (**B**) and (**F**), respectively. (**D, H**) Representative sunflower capitula at the time of transfer to the field. (**D**) LL|28°C capitula do not exhibit coordination of the psuedowhorls or pollen release. (**H**) White ovaries are visible at the margins of the capitula after dissection of post-anthesis florets.

*Figure 7 continued on next page*

*Figure 7 continued*

The online version of this article includes the following figure supplement(s) for figure 7:

**Figure supplement 1.** Pollinator day 2 data.

receptivity is thought to reduce self-pollination and is prevalent throughout the Asteraceae family (*Jeffrey, 2009*).

We found that elongation of ovaries, anther filaments, and styles are all under circadian regulation, but respond differently to environmental cues. For example, transfer to constant light caused a complete inhibition of anther filament elongation, significantly slowed ovary expansion, but had no significant effect on style elongation (*Figure 3—figure supplement 2*). In contrast, transfer to constant darkness slowed style elongation considerably but had only modest effects on growth of ovaries and anther filaments (*Figure 3*; *Figure 3—figure supplement 2*). The strong inhibition of style elongation in constant darkness may help explain why rhythms in style emergence were less robust than those of ovary and anther growth in this condition (*Figure 3*). Responses of the different organs to temperature were also distinct: temperature cycles slowed ovary expansion and had no effect on anther filament growth in either constant darkness or light. In contrast, temperature cycles slowed style elongation rates in constant darkness but not constant light (*Figure 3—figure supplement 2*). These results are consistent with our previous field studies that showed modest changes in ambient temperature affected style but not anther filament elongation (*Creux et al., 2021*). Overall, the different sensitivities of floral organs to light and temperature suggest they may be regulated by distinct growth regulatory pathways. This is consistent with studies in other plants, which revealed that different hormones regulate the growth of male and female reproductive organs (*Chandler, 2011*; *Marciniak and Przedniczek, 2019*).

While the molecular pathways controlling elongation of anther filaments and styles in sunflowers are likely different, we demonstrate that their common regulation by the circadian clock results in the fast and near-synchronous release of pollen a few hours after dawn. Intriguingly, many bee- and butterfly-pollinated Asteraceae species release pollen in the morning (*Budumajji and Raju, 2018*; *Hipólito et al., 2013*; *Neff and Simpson, 1990*; *Valentin-Silva et al., 2016*) while at least one bat-pollinated member of the family releases pollen in the early night (*Amorim et al., 2021*). Since a delay in the timing of pollen release relative to dawn reduces pollinator visits (*Figure 7E–H*) and negatively affects male reproductive success in sunflower (*Creux et al., 2021*), it is tempting to speculate that clock regulation of late-stage floret development may be widespread in the Asteraceae. Identification of the mechanisms by which the circadian system coordinates the growth of male and female floral organs to promote timely pollen release will be of great interest.

The ecological and evolutionary success of the Asteraceae family, estimated to contain ~10% of angiosperm species, is attributed in part to their compressed inflorescences that act together as single false flowers to attract pollinators (*Mandel et al., 2019*). In sunflowers, the individual disk floret primordia are first initiated on the outer perimeter of the very large, flat head meristem and subsequent primordia are initiated following a centripetal pattern to generate the characteristic spiral pattern of immature florets seen across the disk (*Figure 1A*; *Figure 8A*; *Marc and Palmer, 1981*; *Palmer and Steer, 1985*). Recent experimental and modeling studies in gerbera daisies attributed

**Table 3.** Summary of pollinator visits to jet lag and constant light grown sunflowers.

| Experiment | Condition | n | Mean pollen release (hr) | 95% CI pollen release | p-Value | Mean total pollinators | 95% CI total pollinators | p-Value |
|---|---|---|---|---|---|---|---|---|
| LL vs. control | Control A | 7 | 1.16 | ±0.105 | | 104.71 | ±15.6 | |
| LL vs. control | LL\|28°C | 7 | NA | NA | NA | 22.29 | ±5.11 | 2.67E-06 |
| Jet lag Day1 | Control B | 8 | 1.55 | ±0.101 | | 52.38 | ±7.61 | |
| Jet lag Day1 | Jet lag | 11 | 2.89 | ±0.198 | 9.48E-08 | 37.18 | ±8.16 | 0.0456 |
| Jet lag Day 2 | Control B | 7 | 1.73 | ±0.150 | | 33.57 | ±11.7 | |
| Jet lag Day 2 | Jet lag | 11 | 1.39 | ±0.104 | 0.00618 | 55.18 | ±14.0 | NS (0.095) |

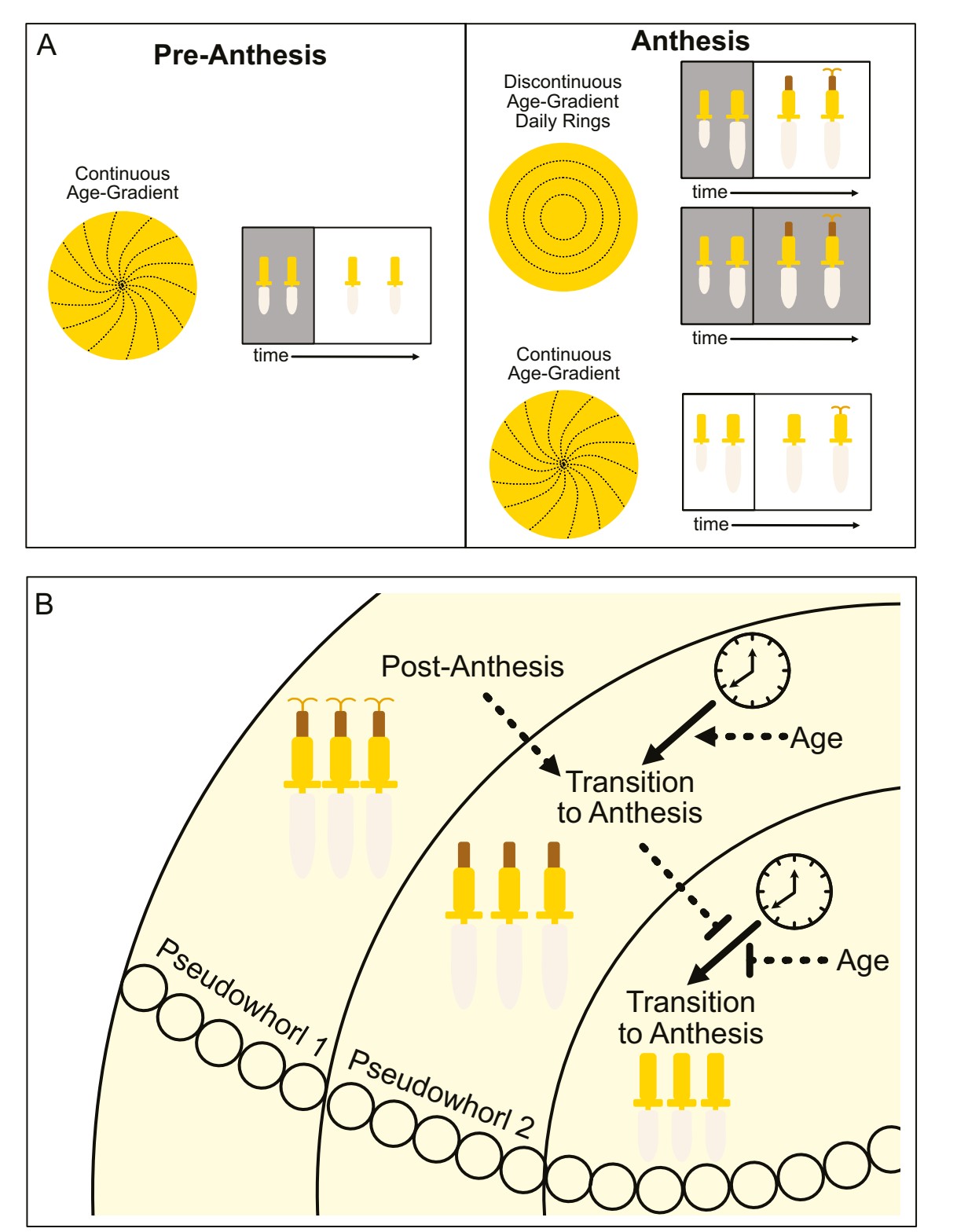

**Figure 8.** Model of sunflower anthesis patterns. (**A**) Developmental patterns observed when sunflowers undergo anthesis in different environmental conditions. Pre-anthesis, the apparent pattern of florets on a capitulum, is a continuous age gradient with spiral phyllotaxy (left panel). During anthesis in both normal light/dark cycles and in constant darkness, the capitulum transitions to a discontinuous age gradient of florets in which rings of florets, or pseudowhorls, undergo synchronized daily rhythms of development (top right panel). In constant light conditions, however, the capitulum maintains

*Figure 8 continued on next page*

*Figure 8 continued*

a continuous age gradient of floret development. Ovaries and styles grow, but stamens do not (bottom right panel). (**B**) Speculative model for transition of floret development from age-dependent continuous gradients to organization into discrete pseudowhorls that undergo coordinated anthesis. After anthesis, florets comprising pseudowhorl 1 send an unknown signal (dashed line) to younger florets, creating a permissive state for the transition to anthesis. In order to respond to this signal, younger florets must both (1) be developmentally competent and (2) be exposed to a sufficient period of darkness during the subjective night (as determined by the circadian clock). Florets younger than pseudowhorl 2 do not elongate floral organs until the post-anthesis state of pseudowhorl 2, floret developmental status, environmental cues, and the circadian clock allow transition to anthesis and the generation of the next pseudowhorl.

formation of this spiral pattern to short-range interactions between spatially close initia (*Zhang et al., 2021*). In contrast, our work suggests that the ring-like pattern of pseudowhorls that emerges during anthesis (*Figure 1E–G*) is generated by the action of the circadian clock on florets across the capitulum disk, with loss of synchronized daily rhythms in anthesis in constant light and temperature leading to loss of pseudowhorl formation (*Figure 6A, B and F*).

How do florets of different developmental ages temporally coordinate anthesis to generate pseudowhorls? We propose that a type of external coincidence model is at work, analogous to the mechanisms underlying photoperiodic control of the transition from vegetative to reproductive growth in many plants (*Pittendrigh and Minis, 1964*; *Yanovsky and Kay, 2003*). In our experiments, developmentally competent immature florets respond to a period of darkness that falls during the subjective night and is at least 4.5 hr long by undergoing the growth events that lead to anthesis (rapid ovary expansion followed by the rapid onset of anther filament and then style elongation) (*Figure 8B*). This is conceptually similar to photoperiodic control of flowering in short day plants such as rice, in which light at night inactivates a clock-regulated transcription factor that promotes the transition to flowering (*Izawa et al., 2002*).

Although the molecular processes underlying light and circadian control of floral organ growth are not currently known, we speculate that they could involve circadian control of transcription coupled with light regulation of protein activity. For example, transcription factors with a nighttime phase of circadian gene expression but that are degraded in the light might promote growth of floral organs. In constant darkness, the activity of these factors would oscillate and drive near-normal anthesis rhythms. In constant light, or if light occurred during the subjective night, the abundance of these proteins would decrease and anthesis would be slow and uncoordinated across the capitulum disk. A similar mechanism has been shown to underlie the circadian and photoperiodic control of hypocotyl elongation in *Arabidopsis* (*Kunihiro et al., 2011*; *Nozue et al., 2007*).

We suggest that local signals may also contribute to the coordination of floret anthesis across the capitulum disk. We found that imposition of dark treatments during the subjective day delayed anthesis by close to 24 hr (*Figure 4*; *Figure 4—figure supplement 1J*), at which point normal rhythmic patterns of anthesis resumed after entrainment to the new light conditions. However, pseudowhorls that developed after these delays contained the same number of florets as those found on plants that did not experience delays (*Figure 4—figure supplement 2*). This suggests that signaling may occur between florets in adjacent pseudowhorls to control the onset of anthesis. This might be a positive cue from post-anthesis florets to promote development of the next pseudowhorl, or a negative cue from pre-anthesis florets to inhibit development of nearby florets (*Figure 8B*).

Plants are generally considered diurnal organisms and often have more robust circadian rhythms in constant light than in constant darkness (*Michael et al., 2008*). We were therefore surprised to find robust rhythms of anthesis in conditions of constant temperature and darkness but not constant temperature and light (*Figure 3*; *Figure 6*). Similar observations have been made in *Neurospora crassa* and in many animals, with circadian rhythms observed in constant darkness but not in constant light (*Aschoff, 1979*; *Sargent et al., 1966*). An interesting question for further study is whether in constant light molecular rhythms are present in sunflower florets but developmental rhythms are masked, or whether the floral circadian clock is arrhythmic in this condition. The very rapid resetting of the different phases of organ growth to new light conditions (*Figure 4—figure supplement 1*) suggests the latter may be true.

Another interesting question is why florets of different developmental ages undergo coordinated anthesis to generate pseudowhorls. It was reported that the foraging activity of both honeybees and native bees on sunflower heads is positively correlated with the number of florets undergoing anthesis

at any given time, and that seed set is strongly correlated with the number of bees foraging on heads (*DeGrandi-Hoffman and Watkins, 2000*; *Neff and Simpson, 1990*). In the case of the highly social honeybees, hive members communicate via a dance language to quickly reallocate the numbers of foragers sent to different food sources based on resource availability (*Visscher and Seeley, 1982*). Plants treated with constant light so that floret anthesis was not coordinated across the capitulum disk were much less attractive to pollinating insects than were control plants (*Figure 7A–D*), presumably due to the lack of pollen release and/or the uncoordinated development of florets across the capitulum disk. We speculate that both factors have an effect, and that the coordinated release of floral rewards by the hundreds of florets undergoing anthesis within a pseudowhorl each morning may be a mechanism to enhance attractiveness to pollinators and thus promote reproductive success.

## Materials and methods

**Key resources table**

| Reagent type (species) or resource | Designation | Source or reference | Identifiers | Additional information |
|---|---|---|---|---|
| Genetic reagent (*Helianthus annuus*) | HA412 HO | US Department of Agriculture, US National Plant Germplasm System | PI 603993 | https://npgsweb.ars-grin.gov/gringlobal/search |
| Other | Raspberry pi cameras and computers | Raspberry Pi | NoIR V2 cameras, 3 Model B computers | low cost, small computers and cameras used for time-course image capture |
| Other | Wingscapes Timelapse Cam | https://www.wingscapes.com | WCT-00121 | weatherproof camera used for time-course image capture |
| Software | R, base package | https://www.r-project.org | | |
| Software | R, tidyverse package | https://cran.r-project.org/web/packages/tidyverse/index.html | RRID:SCR_019186 | |
| Software | R, circular package | https://cran.r-project.org/web/packages/circular/index.html | RRID:SCR_003005 | |
| Software | BioDare2 | https://biodare2.ed.ac.uk | | |
| Software | ImageJ software | ImageJ (http://imagej.nih.gov/ij/) | RRID:SCR_003070 | |

### Sunflower floret development imaging and scoring

Sunflower seeds of HA412 HO (USDA ID: PI 603993) genotype were planted into small pots of soil (Sunshine Mix #1, Sun Gro Horticulture) and germinated with a plastic lid in a PGV36 growth chamber (Conviron, Winnipeg, MB, Canada) at 25°C with 16 hr light (provided by metal halide and incandescent lamps, 300 µmol m$^{-2}$ s$^{-1}$) and 8 hr darkness per day. Plants were watered with nutrient water containing an N-P-K macronutrient ratio of 2:1:2. Two weeks after sowing, seedlings were transplanted to 2-gallon pots with 1 scoop of Osmocote fertilizer (SMG Brands). Approximately 60 days after sowing, sunflower capitula entering anthesis in their first pseudowhorl were transferred to an PGR15 growth chamber (Conviron, Winnipeg, MB, Canada; 200 µmol m$^{-2}$ s$^{-1}$ provided by metal halide and incandescent lamps) to image floret development under the indicated environmental conditions. Transfer to experimental conditions occurred at ZT 0. Sunflower stalks and capitula were taped to bamboo stakes (to avoid capitulum moving out of the camera frame). Raspberry Pi NoIR V2 cameras were mounted on Raspberry Pi 3 model B computers (Raspberry Pi, Cambridge, UK); cameras were fitted with LEE 87 75×75 mm$^2$ infrared (IR) filters (Lee Filters, Andover, England). Computers were programmed to take a photo every 15 min. Infrared LEDs (Mouser Electronics, El Cajon, CA, USA) were programmed to flash during image capture so that the capitula were visible in the dark without disrupting plant growth. Sunflowers were imaged immediately upon transfer to experimental conditions, and through anthesis of all florets in the capitulum. The 15 min interval images were analyzed sequentially in a stack on ImageJ (*Schneider et al., 2012*). For each image, the ovaries, stamens, and styles were scored for a change in size from the previous image. Ovary growth was seen as corolla tube swelling above the immature capitulum surface (*Figure 1E-G*). Late-stage anther filament elongation was observed starting from when the corolla tube cracked open to reveal the stamen tube

until its full extension above the corolla surface (*Figure 1D* – II, 1G). Late-stage style elongation was observed starting with the visible extrusion of pollen out of the top of the anther tube and ending with the style fully extended (*Figure 1D* – III, 1G). Organs were classified as growing when >5% of the florets in a pseudowhorl showed a change in length in one time-lapse image relative to the previous one. Growth was measured qualitatively as active or inactive.

## Polar plots and statistical analyses

All sunflower scoring data was plotted in R using the tidyverse package (*Wickham et al., 2019*). To create polar plots based on each pseudowhorl, scoring data was separated into pseudowhorls. At times, more than 24 hr of data is presented on one polar plot. In the environmental conditions that eliminated pseudowhorl coordination, polar plots were created based on 24 hr of data. The average time of anthesis for each organ and standard deviation was calculated using the circular package in R (*Agostinelli and Lund, 2022*). This information was calculated for each pseudowhorl, all pseudowhorls together superimposed on one 24 hr cycle, and for pseudowhorls 3 and up (after resetting is complete). The BioDare2 website (https://biodare2.ed.ac.uk/) was used to calculate rhythmic parameters for all conditions and organs (*Zielinski et al., 2014*). The FFT-NLLS fit method was used to calculate period and phase values based on average peak values. The time frame analyzed was different in each condition and is reported in *Table 1*.

## Scoring consensus growth for florets in a parastichy

The 15 min interval images were analyzed sequentially in a stack on ImageJ. A clockwise parastichy was selected, and florets numbered from the outer edge of the pseudowhorl inward. Images taken at 15 min intervals, the ovaries, stamens, and styles for each floret were scored for a change in size from the previous image. Four parastichies per condition were analyzed, from two different sunflower heads. In LL|25°C conditions, which had no pseudowhorls, 10 florets in a parastichy were scored for 48 hr. For all other conditions tested, a parastichy from the second pseudowhorl was scored until anthesis of all florets per parastichy within the pseudowhorl was complete. For the DD|25°C condition, the fourth pseudowhorl was also scored.

## Counting florets per pseudowhorl

For the 4.5 hr dark conditions imposed at various times of day, the number of florets that began anthesis per pseudowhorl was counted every 24 hr. For each condition, florets from 12 to 19 parastichies from two different sunflower heads were counted.

## Floret organ timecourse and ovary growth measurements

To measure the organs in florets over a timecourse (*Figure 2A*; LD|25°C [19.5 hr:4.5 hr]) entire florets from a single pseudowhorl were dissected from the capitulum and photographed. Organs were then measured with ImageJ for each floret. For ovary growth measurements, at ZT 2 on after the second day in a given growth condition, entire florets, including ovaries, were dissected from the capitulum along four to eight continuous parastichies and photographed. Ovary length was then measured with ImageJ for each floret, consecutively. Position 8 always designated the boundary between pseudowhorls.

## Scoring of pollinator visits

These experiments were conducted in Davis, CA, June-August, 2022. Seeds were germinated and grown for 3 weeks in a PGV36 growth chambers (Conviron, Winnipeg, MB, Canada) as described above, at which point they were transferred to a field site. Before the start of anthesis plants were transferred to either a PGV36 or a PGR15 growth chamber (Conviron, Winnipeg, MB, Canada) for either 2 days for LL|28°C and Control A plants, or 1 week for jet lag and Control B plants. Controls were maintained in chambers with light, temperature, and humidity cycling in coordination with the local average daily forecast for the following week. Constant light plants were maintained at a constant 28°C, with 300 μmol m$^{-2}$ s$^{-1}$ provided by metal halide and incandescent lamps. Jet lagged plants were entrained to the same conditions as the control plants but with a 3 hr phase delay. Plants were transferred to the field just before anthesis of the second or third pseudowhorl, just after dawn in the case of the constant light experiment and 1 hour before dawn for the jet lag experiment. In the

jet lag experiments, florets that had opened on previous days were removed with forceps to limit the pollinator recruitment signals to florets developing on the day of the experiment. Pots were arranged so the capitula faced east and stems were secured to bamboo poles for imaging. Images were taken at 5 min intervals using BirdCam 2.0 cameras (Wingscapes). Pollinator visits were scored using Image-Glass (https://imageglass.org/). Any insect large enough to be seen in the images was counted. New visits were counted when an insect landed on or changed location on the disk florets. The first image in which pollen was visible on the tip of the stamen was scored for time of pollen presentation. All data was plotted in R using ggplot (*Wickham, 2016*) and tidyverse (*Wickham et al., 2019*).

## Acknowledgements

This work was supported by grants from the National Science Foundation (IOS 1759942) and the US Department of Agriculture-National Institute of Food and Agriculture (CA-D-PLB-2259-H) to SLH. We thank Chris Brooks, Hongtao Zhang, and Cassandra Baker for help tending to and running experiments on sunflowers; Julin Maloof and John Davis for helpful discussions and statistical and scripting advice; and Mike Covington for his R script to generate polar plots.

## Additional information

### Funding

| Funder | Grant reference number | Author |
| --- | --- | --- |
| National Science Foundation | IOS 1759942 | Stacey L Harmer |
| U.S. Department of Agriculture | CA-D-PLB-2259-H | Stacey L Harmer |

The funders had no role in study design, data collection and interpretation, or the decision to submit the work for publication.

### Author contributions

Carine M Marshall, Conceptualization, Formal analysis, Investigation, Methodology, Writing - original draft, Writing - review and editing; Veronica L Thompson, Investigation, Writing - review and editing; Nicky M Creux, Conceptualization, Methodology, Writing - review and editing; Stacey L Harmer, Conceptualization, Supervision, Funding acquisition, Writing - original draft, Writing - review and editing

### Author ORCIDs

Carine M Marshall http://orcid.org/0000-0001-7178-4664
Veronica L Thompson http://orcid.org/0000-0003-0500-5639
Nicky M Creux http://orcid.org/0000-0002-4179-6995
Stacey L Harmer http://orcid.org/0000-0001-6813-6682

### Decision letter and Author response

Decision letter https://doi.org/10.7554/eLife.80984.sa1
Author response https://doi.org/10.7554/eLife.80984.sa2

## Additional files

### Supplementary files
• MDAR checklist

### Data availability

All source data have been uploaded to Dryad under the following accession codes: 10.25338/B8865X (timelapse scoring), 10.25338/B86358 (pollinator visits), 10.25338/B8963G (consensus scoring), 10.25338/B8CW5R (ovary measurements), and 10.25338/B8HP9F (organ growth kinetics).

The following datasets were generated:

| Author(s) | Year | Dataset title | Dataset URL | Database and Identifier |
|---|---|---|---|---|
| Marshall C, Creux N | 2023 | Sunflower timelapse scoring | https://doi.org/10.25338/B8865X | Dryad Digital Repository, 10.25338/B8865X |
| Marshall C, Thompson V | 2022 | Sunflower pollinator visit scoring | https://doi.org/10.25338/B86358 | Dryad Digital Repository, 10.25338/B86358 |
| Marshall C | 2023 | Sunflower consensus scoring | https://doi.org/10.25338/B8963G | Dryad Digital Repository, 10.25338/B8963G |
| Marshall C | 2023 | Sunflower ovary measurements | https://doi.org/10.25338/B8CW5R | Dryad Digital Repository, 10.25338/B8CW5R |
| Marshall C, Thompson V | 2022 | Organ kinetics measurements | https://doi.org/10.25338/B8HP9F | Dryad Digital Repository, 10.25338/B8HP9F |

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
