## [Editor Report]

This study successfully illustrates how the circadian clock spatiotemporally regulates unique developmental patterns of sunflower anthesis. It is generally true that the circadian clock underlies nearly all developmental processes in plants, but it could be surprising that the exquisite changes in flower shapes and growth in a disc structure are also shaped by the circadian clock. They also nicely put the emphasis on the relevance of the anthesis patterns in the ecological context in that the spatiotemporal coordination of flowering is essential for pollination and reproductive fitness.

---

## [Decision Letter]

**Decision letter after peer review:**

Thank you for submitting your article "The circadian clock controls temporal and spatial patterns of floral development in sunflower" for consideration by *eLife*. Your article has been reviewed by 3 peer reviewers, including Pil Joon Seo as Reviewing Editor and Reviewer #1, and the evaluation has been overseen by Jürgen Kleine-Vehn as the Senior Editor.

Essential revisions:

Two critical issues should be addressed.

(1) Molecular mechanism underlying circadian control of floral development should be proposed. Instead of rigorous genetic and biochemical analyses, reviewers want to see possible scenario behind the interesting observations with a potential linkage of certain circadian clock components.

(2) Ecological relevance of the spatiotemporally coordinated floral development should be addressed. The characteristic floral development patterns would be related to evolutionary advantages, such as enhanced plant fitness and reproductive success.

*Reviewer #1 (Recommendations for the authors):*

The study by Marshall et al. demonstrates the role of the circadian clock in spatiotemporal regulation of floral development. The authors nicely illustrated floral development patterns in domesticated sunflower. In particular, during anthesis, discrete developmental zones, namely pseudowhorls, are established, and hundreds of florets simultaneously undergo maturation in each psudowhorl, which is under the control of the circadian clock. The flower development is temperature-compensated and also involves circadian gating of light/dark response. Overall, even though the mechanistic understanding remains obscure, circadian regulation of floral development in sunflower is clearly shown, which may fit the readership of *eLife*. A few concerns should be addressed before the publication.

1. Is circadian rhythm in flower synchronized systemically with whole-plant rhythm? Otherwise, flower-specific circadian system might be segmented in controlling floral development. This reviewer is also curious if the circadian actions can be distinguished between pseudowhorl and pre-anthesis capitulum.

2. Expression of core circadian clock genes could be analyzed in several subregions of flowers, including pseudowhorl 1, 2 and remaining parts.

3. If any circadian mutant of sunflower is available, I would like to recommend authors to analyze whether the circadian mutant display defects in floral development patterns, such as anthesis and pseudowhorl formation.

*Reviewer #2 (Recommendations for the authors):*

The authors carefully measured the timing of elongation of ovary, stamen, and style in the sunflower capitulum and found that the circadian clock regulates floral development. All results are clearly described, and the main take-home message of this paper is clear as well.

In addition, at the end of this paper, the authors suggest two interesting hypotheses, which are related to (1) the origin of the clock signal and (2) the role of the circadian clock in plant fitness. Although the authors manipulated the light and temperature conditions to examine whether the circadian clock regulates floral development, the depth and strength of the evidence in this paper are not enough for publishing in *eLife*. There are no molecular evidence and ecological experiments in this paper. It will be very interesting if the authors provide any evidence to falsify the two hypotheses presented at the end of the paper.

In 2016, the same group published a very interesting story of sunflowers in Science, which examined how the clock regulates sunflower heliotropism in a molecular level and measured the ecological performance of the clock-altered flowers with a proper pollinator. The major trait that the authors examined in this paper is the elongation of cells. Several molecular components are known to be involved in cell elongation, such as cell wall modification enzymes and osmotic pressure regulators. It could be interesting whether the diurnal rhythm of floral development in each organ is correlated with the transcript levels of cell wall modification enzymes in the organ. Time-series experiments of RNA sequencing can be performed easily these days.

*Reviewer #3 (Recommendations for the authors):*

Overall this is a strong article and in the opinion of this reviewer, further experimentation is not required in order to secure publication. I have a set of specific recommendations that I hope the authors find useful to improve their study:

1. Please consider alternative interpretations of the coincidence model, as suggested in the "public review" section of my review.

2. Plots of the temporal clustering of specific developmental processes are used frequently in the study (e.g. Figure 2B, Figure 3A, etc.). In these plots, I was puzzled by how the colored oval that surrounds the data points was calculated. Do the x axis limits of these ovals represent a confidence interval within the data? Please can some further explanation or justification for these ovals be presented within the legends of relevant figures.

3. In the section from lines 127-152, please set out specifically which mechanisms you consider to be continuous development and which you consider to be discrete developmental processes. This is suggested by the title of the subsection, so it would be helpful to link each paragraph to one of these ideas a little more explicitly (e.g., "Therefore, this subset of developmental processes can be considered continuous…."). I think it could help an unfamiliar reader.

4. Within lines 156-168, the authors discuss anther filament length. This does not appear to be measured in Figure 2A. Is filament length inferred from anther tube length? This reviewer is not familiar with the botanical detail of flower development, so it would be helpful for readers in a similar position to have a little more explanation of this in this section of the results.

5. The phase stability of each of the developmental processes under different photoperiods provides strong evidence for circadian regulation. The authors might have missed the opportunity to point that out to readers.

6. On lines 269-270, the authors state that entrain might occur quickly in the capitula, without any point of reference to other systems. I am not convinced it is any faster than for Arabidopsis, but perhaps a broader comparison could be helpful to back up this claim.

7. On lines 426-428, the authors discuss their findings in relation to light conditions that permit circadian rhythms. There are examples beyond Neurospora of rhythms that are conditional upon constant darkness (e.g. examples in *Drosophila*, rodents) that are worth mentioning. I tend to agree that this is unusual for plants- though there are some unusual examples (the rhythm of delayed chlorophyll fluorescence in wheat, from Anthony Hall's lab, being one example).

---

## [Author Response]

Essential revisions:Two critical issues should be addressed.(1) Molecular mechanism underlying circadian control of floral development should be proposed. Instead of rigorous genetic and biochemical analyses, reviewers want to see possible scenario behind the interesting observations with a potential linkage of certain circadian clock components.

We have added a paragraph to the Discussion section suggesting possible molecular mechanisms underlying our observations that floret organ development is regulated both by the circadian clock and light signaling pathways. We draw analogies with Arabidopsis PIF4 and PIF5 transcription factors, which regulate elongation growth of hypocotyls and petioles. Transcription of these factors is regulated by the circadian clock, while their protein stability is regulated by light. A similar mechanism may underlie circadian and light regulation of the elongation of ovaries, anther filaments, and styles in sunflower.

(2) Ecological relevance of the spatiotemporally coordinated floral development should be addressed. The characteristic floral development patterns would be related to evolutionary advantages, such as enhanced plant fitness and reproductive success.

We have conducted two new experiments to better frame our observations in an ecological context (Figure 7 and Figure 7-S1). First, we investigated the attractiveness of sunflowers maintained in constant light conditions to pollinating insects. We found that growth in constant light, which disrupts the synchronous development of florets to form pseudowhorls, led to flowers that were much less attractive to pollinators than those grown in normal light/dark cycles. Next, we examined the importance of the timing of anthesis. We found that plants entrained so that anthesis was delayed several hours relative to dawn received significantly fewer insect visits than control plants. Together, these experiments demonstrate that appropriate circadian and environmental regulation of floral development are key to optimize visits by pollinating insects and are thus implicated in male reproductive fitness.

Reviewer #1 (Recommendations for the authors):The study by Marshall et al. demonstrates the role of the circadian clock in spatiotemporal regulation of floral development. The authors nicely illustrated floral development patterns in domesticated sunflower. In particular, during anthesis, discrete developmental zones, namely pseudowhorls, are established, and hundreds of florets simultaneously undergo maturation in each psudowhorl, which is under the control of the circadian clock. The flower development is temperature-compensated and also involves circadian gating of light/dark response. Overall, even though the mechanistic understanding remains obscure, circadian regulation of floral development in sunflower is clearly shown, which may fit the readership of eLife. A few concerns should be addressed before the publication.1. Is circadian rhythm in flower synchronized systemically with whole-plant rhythm? Otherwise, flower-specific circadian system might be segmented in controlling floral development. This reviewer is also curious if the circadian actions can be distinguished between pseudowhorl and pre-anthesis capitulum.2. Expression of core circadian clock genes could be analyzed in several subregions of flowers, including pseudowhorl 1, 2 and remaining parts.

We agree that these are interesting questions but believe they would be best addressed in a future study.

3. If any circadian mutant of sunflower is available, I would like to recommend authors to analyze whether the circadian mutant display defects in floral development patterns, such as anthesis and pseudowhorl formation.

Unfortunately, we do not have any sunflower mutants with circadian phenotypes. We have attempted to generate transgenic plants in the past but with little success.

Reviewer #2 (Recommendations for the authors):The authors carefully measured the timing of elongation of ovary, stamen, and style in the sunflower capitulum and found that the circadian clock regulates floral development. All results are clearly described, and the main take-home message of this paper is clear as well.In addition, at the end of this paper, the authors suggest two interesting hypotheses, which are related to (1) the origin of the clock signal and (2) the role of the circadian clock in plant fitness. Although the authors manipulated the light and temperature conditions to examine whether the circadian clock regulates floral development, the depth and strength of the evidence in this paper are not enough for publishing in eLife. There are no molecular evidence and ecological experiments in this paper. It will be very interesting if the authors provide any evidence to falsify the two hypotheses presented at the end of the paper.

As noted in our response to Reviewer #1, we have performed new experiments that suggest that circadian and light regulation of floral anthesis may have significant impacts on male reproductive fitness.

In 2016, the same group published a very interesting story of sunflowers in Science, which examined how the clock regulates sunflower heliotropism in a molecular level and measured the ecological performance of the clock-altered flowers with a proper pollinator. The major trait that the authors examined in this paper is the elongation of cells. Several molecular components are known to be involved in cell elongation, such as cell wall modification enzymes and osmotic pressure regulators. It could be interesting whether the diurnal rhythm of floral development in each organ is correlated with the transcript levels of cell wall modification enzymes in the organ. Time-series experiments of RNA sequencing can be performed easily these days.

We agree that these are interesting questions but believe they would be best addressed in a future study.

Reviewer #3 (Recommendations for the authors):Overall this is a strong article and in the opinion of this reviewer, further experimentation is not required in order to secure publication. I have a set of specific recommendations that I hope the authors find useful to improve their study:

We thank the reviewer for their enthusiasm and constructive comments.

1. Please consider alternative interpretations of the coincidence model, as suggested in the "public review" section of my review.

We thank the reviewer for their positive comments and overall enthusiasm for the study. We agree that it is entirely plausible that continuous light masks circadian clock-controlled rhythms in floral organ development; in our view, this is a restatement of the external coincidence model. We argue that in developing sunflowers, a circadian clock-regulated process controls elongation of floret organs. Normal development depends upon a dark period of at least 4.5 hours occurring during the subjective night. In constant light conditions, or early in reentrainment when the dark period occurs during the subjective day, normal development is inhibited. This model is analogous to the photoperiodic control of flowering time in short-day plants, in which light perceived during the subjective night inhibits the floral transition.

2. Plots of the temporal clustering of specific developmental processes are used frequently in the study (e.g. Figure 2B, Figure 3A, etc.). In these plots, I was puzzled by how the colored oval that surrounds the data points was calculated. Do the x axis limits of these ovals represent a confidence interval within the data? Please can some further explanation or justification for these ovals be presented within the legends of relevant figures.

The description in the legend of Figure 2 has been expanded. The colored ovals were generated from the same data points that are also plotted as white points in the figures. The colored regions do not convey statistical significance and are merely intended to allow viewers to easily identify florets belonging to separate pseudowhorls.

3. In the section from lines 127-152, please set out specifically which mechanisms you consider to be continuous development and which you consider to be discrete developmental processes. This is suggested by the title of the subsection, so it would be helpful to link each paragraph to one of these ideas a little more explicitly (e.g., "Therefore, this subset of developmental processes can be considered continuous…."). I think it could help an unfamiliar reader.

We thank the reviewer for the suggestion and have expanded our description of what we consider continuous and discrete developmental patterns in the first paragraph of the Results section.

4. Within lines 156-168, the authors discuss anther filament length. This does not appear to be measured in Figure 2A. Is filament length inferred from anther tube length? This reviewer is not familiar with the botanical detail of flower development, so it would be helpful for readers in a similar position to have a little more explanation of this in this section of the results.

In Figure 2 and its legend we define stamens as filament plus anther tube and in Figure 2A we show that stamens elongate but anther tubes do not. We conclude that stamen protrusion is controlled by filament elongation, consistent with previous reports (Baroncelli et al., 1990; Lobello et al., 2000). We have now stated this explicitly (line 184).

5. The phase stability of each of the developmental processes under different photoperiods provides strong evidence for circadian regulation. The authors might have missed the opportunity to point that out to readers.

We thank the reviewer for the suggestion and have added a sentence to describe this (lines 209-211).

6. On lines 269-270, the authors state that entrain might occur quickly in the capitula, without any point of reference to other systems. I am not convinced it is any faster than for Arabidopsis, but perhaps a broader comparison could be helpful to back up this claim.

We did not suggest that re-entrainment occurs more rapidly in sunflower capitula than in other plant systems. We feel an in-depth discussion of relative rates of entrainment would detract from the flow of the manuscript so have opted not to discuss it further.

7. On lines 426-428, the authors discuss their findings in relation to light conditions that permit circadian rhythms. There are examples beyond Neurospora of rhythms that are conditional upon constant darkness (e.g. examples in *Drosophila*, rodents) that are worth mentioning. I tend to agree that this is unusual for plants- though there are some unusual examples (the rhythm of delayed chlorophyll fluorescence in wheat, from Anthony Hall's lab, being one example).

We thank the reviewer for these comments and have noted that a number of animals are also rhythmic in constant darkness but arrhythmic in constant light (lines 570-576).